# PASS: Private Attributes Protection with Stochastic Data Substitution

**Yizhuo Chen** [1 2]  **Chun-Fu (Richard) Chen** [2]  **Hsiang Hsu** [2]  **Shaohan Hu** [2]  **Tarek Abdelzaher** [1]

## Abstract

The growing Machine Learning (ML) services require extensive collections of user data, which may inadvertently include people's private information irrelevant to the services. Various studies have been proposed to protect private attributes by removing them from the data while maintaining the utilities of the data for downstream tasks. Nevertheless, as we theoretically and empirically show in the paper, these methods reveal severe vulnerability because of a common weakness rooted in their adversarial training based strategies. To overcome this limitation, we propose a novel approach, PASS, designed to stochastically substitute the original sample with another one according to certain probabilities, which is trained with a novel loss function soundly derived from information-theoretic objective defined for utility-preserving private attributes protection. The comprehensive evaluation of PASS on various datasets of different modalities, including facial images, human activity sensory signals, and voice recording datasets, substantiates PASS's effectiveness and generalizability.

## 1. Introduction

The expansion of modern Machine Learning (ML) services has seamlessly improved the convenience of daily lives, where the service providers oftentimes gather data from users and then utilize advanced models to cater to users' needs (Secinaro et al., 2021; Ahmed et al., 2022). However, the data collected frequently includes private information that users may be reluctant to disclose (Boulemtafes et al., 2020; Kumar et al., 2023). For instance, a human voice recognition service needs to collect speaker's voice

to extract the content (Kheddar et al., 2024), which would inadvertently contain private attributes such as the speaker's gender or accent, imposing the risk of privacy leakage when an adversary tries to eavesdrop on the voice. Therefore, there has been a lasting interest in developing a data obfuscation module that can be inserted into the data sharing or ML service pipelines to protect the private attributes by suppressing or removing them from the data, while maintaining the utility of the data for downstream tasks.

Various methods are proposed towards this goal. Roy & Boddeti (2019); Bertran et al. (2019); Wu et al. (2020) focused on suppressing private attributes while preserving explicitly annotated useful attributes. On the other hand, Huang et al. (2018); Malekzadeh et al. (2019); Dave et al. (2022) extended their researches to managing unannotated general features for broader applicability. Furthermore, Chen et al. (2024) summarized and satisfied *SUIFT*, 5 desirable properties of utility-preserving private attributes protection methods. Notably, these state-of-the-art methods are adversarial training based, where their obfuscation module is trained to prevent an adversarial private attributes classifier from making correct inferences.

However, as widely discussed in the neighboring fields of adversarial robustness (Carlini & Wagner, 2017; Athalye et al., 2018; Ilyas et al., 2019; Carlini et al., 2019), generated image/vedio detection (Yu et al., 2019; Wang et al., 2020; Masood et al., 2023), membership inference attacks (Carlini et al., 2022) and gradient inversion in federated learning (Geiping et al., 2020; Huang et al., 2021), a common weakness with adversarial training based methods is that, although the defender can tolerate the jointly-trained adversary, it may be vulnerable to slightly stronger or unseen adversaries. In the private attributes protection context, as we theoretically and empirically demonstrate in Section 3.2, this weakness can indeed cause significant negative impact on the state-of-the-art methods, reducing their practicality in real-world deployment.

To overcome the above weakness, we present a novel method for private attributes protection, called **P**rivate **A**ttributes protection with **S**tochatsic data **S**ubstitution (or **PASS**), which avoids the adversarial training strategy altogether. Specifically, we propose to substitute each input sample of the data sharing system or ML service pipeline with another sample according to a stochastic data substitu-

---

[1]Department of Computer Science, University of Illinois Urbana-Champaign, USA [2]Global Technology Applied Research, JPMorgan Chase, USA. Correspondence to: Yizhuo Chen <yizhuoc@illinois.edu>, Chun-Fu (Richard) Chen <richard.cf.chen@jpmchase.com>.

*Proceedings of the $42^{nd}$ International Conference on Machine Learning*, Vancouver, Canada. PMLR 267, 2025. Copyright 2025 by the author(s).

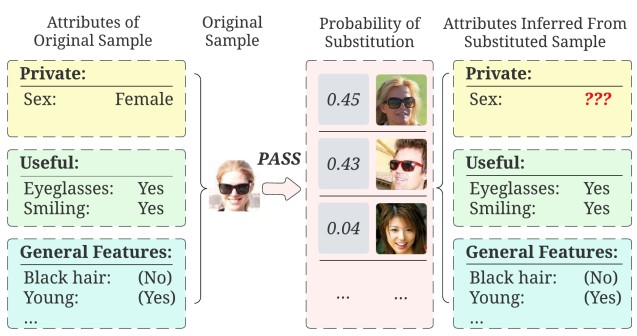

Figure 1. An illustration use case of PASS applied on facial images. We suppress "sex" as a private attribute, and preserve "Eyeglasses" and "Smiling" as useful attributes. Apart from these attributes, we also preserve general features in facial images, such as "Black hair" and "Young", which are not explicitly annotated in the dataset, but are useful for potential downstream applications. PASS stochastically substitutes the original sample with another sample such that the private attribute cannot be accurately inferred from the substituted sample. On the contrary, the useful attributes and general features are still inferable from the substituted sample.

tion algorithm. This algorithm is parameterized by a neural network and is trained with our novel loss function derived step-by-step from an information-theoretic objective defined for utility-preserving private attributes protection. Benefiting from the theoretical basis, PASS has clear operational boundaries for entangled attributes and can trade-off between privacy and utility controllably. An illustrative use case of PASS is provided in Figure 1.

In summary, our paper's contributions are threefold: 1) We propose PASS, a stochastic data substitution based method that overcomes the common weakness of state-of-the-art private attributes protection methods; 2) We demonstrate theoretically that PASS is rigorously rooted in information theory with desirable properties; and 3) We extensively evaluate PASS on facial images, human activity sensory signals, and human voice recordings to show its broad applicability.

## 2. Related Works

**Utility-preserving Private Attributes Protection.** Various studies have been proposed on this topic recently. AttriGuard(Jia & Gong, 2018) proposed to defend against attribute inference attacks using evasion attack on the adversarial classifier. PPDAR(Wu et al., 2020), GAP(Huang et al., 2018) and MaSS(Chen et al., 2024) proposed to minimize private information leakage by making the adversarial classifier unable to make correct predictions. In comparison, ALR(Bertran et al., 2019) and BDQ(Kumawat & Nagahara, 2022) proposed to minimize private information leakage by making the adversarial classifier uncertain about its predictions. Maxent-ARL(Roy & Boddeti, 2019) and MSDA(Malekzadeh et al., 2019) adopted a combination of

both above strategies. Different from above, SPAct(Dave et al., 2022) proposed to train the obfuscation model to maximize a contrastive learning loss adversarially with its feature extractor. All of the above methods, except for GAP, proposed to preserve annotated useful attributes by ensuring their predictability. Generalizing the problem, GAP, MSDA, MaSS, and SPAct(Dave et al., 2022) proposed to manage unannotated general features. These works are adversarial training based, resulting in a common weakness elaborated in Section 3.2, motivating the design of PASS.

The field of fairness shares similar problem formulation and designing techniques with private attributes protection (Edwards & Storkey, 2016; Madras et al., 2018; Sarhan et al., 2020; Caton & Haas, 2024). Nevertheless, their training losses are typically derived from fairness metrics, as opposed to the privacy objectives, making them uncomparable with private attributes protection methods.

**Local Differential Privacy (LDP) And Randomized Response.** Differential Privacy requires the randomized mechanism to produce similar distributions when applied to any two neighboring datasets (Dwork et al., 2006), which is typically achieved with additive noises (Dwork et al., 2014; Abadi et al., 2016; Zhang et al., 2018; Sun et al., 2020). Similarly, Local Differential Privacy casts the same requirement on any two samples in a dataset (Kasiviswanathan et al., 2011; Yang et al., 2023; Arachchige et al., 2019), instead of two neighboring datasets. Local Differential Privacy is mainly achieved with Randomized Response (Warner, 1965; Wang et al., 2016; Chaudhuri & Mukerjee, 2020), which typically involves randomly switching the class of the sensitive attribute when collected. As shown in Appendix B, under certain assumptions, PASS can be viewed as a LDP mechanism—specifically, an extension of randomized response, that is distinguished by its utility preservation requirements and applicability on high-dimensional space.

**k-Anonymity, l-Diversity and t-Closeness (k-l-t privacy).** PASS can also be viewed as an extension to the k-l-t privacy in high-dimensional spaces. k-anonymity aims at thwarting membership inference by ensuring that at least k samples share the same identifiable attributes after obfuscation, and consequently undistinguishable (Samarati & Sweeney, 1998; Qu et al., 2017; Song et al., 2019), while PASS aims at thwarting private attribute inference by ensuring that multiple samples with different private attributes are substituted by each other, and consequently also undistinguishable. l-diversity and t-closeness extended k-anonymity, requiring that these k-samples (equivalent group) have diverse private attributes (Li et al., 2006), preferably with similar distribution as the entire dataset (Sei et al., 2017; Rajendran et al., 2017; Majeed & Lee, 2020), preventing the attacker from guessing the private attribute from obfuscated sample accurately. Similarly, PASS also encourages the private attribute

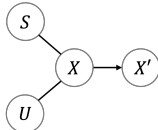

Figure 2. The probabilistic model of all random variables. $U, S, X'$ are only dependent on $X$.

guessed from the substituted sample to have a similar distribution as in the entire dataset.

## 3. Problem Definition and Motivation

We aim to build a data obfuscation algorithm that could be inserted into a data sharing or processing pipeline to remove certain private attributes from an input sample while preserving its useful attributes as well as general features for downstream tasks. To ensure practicality and soundness, we adhere to the restrictive *SUIFT* requirements as recently proposed in Chen et al. (2024), as elaborated below.

**Sensitivity suppression** and **Utility preservation** are the most basic requirements, suggesting that our method should ensure that private attributes are no longer predictable after obfuscation while the specified useful attributes are maintained. For example, when sharing human voices, the user of our method may remove the speaker's gender information from the audio clip, while preserving the spoken content.

**Invariance of sample space** dictating that the obfuscated data should remain in the same space as the original data, ensuring the seamless insertion of our method into existing pipelines and data re-usability, which is adopted in most of the recent works (Bertran et al., 2019; Dave et al., 2022; Chen et al., 2024; Malekzadeh et al., 2019).

**Feature management without annotation** requires preserving the unspecified general features in the data. In the human voice example, general features may include the speaker's accent and age, etc., which are not explicitly specified or labeled in the dataset. This requirement ensures broader usage of our method in the real world, where the downstream tasks are oftentimes unknown, plentiful, and constantly evolving (Huang et al., 2018; Dave et al., 2022; Chen et al., 2024).

Finally, **Theoretical basis** is also required for the entire proposed framework to enhance soundness and correctness.

### 3.1. Information-theoretic Problem Definition for Private Attributes Protection

Following the high-level requirements, we formulate our problem into an information-theoretic framework for in-depth understanding. For presentation clarity, we use the following notation convention in our paper: uppercase letters (e.g., $X, S$) denote random variables, and their corresponding lowercase letters (e.g., $x, s$) the realization of random variables. We use $P(\cdot)$ to denote probability distributions (e.g., $P(X)$), among which we use $P_{\text{data}}(\cdot)$ to indicate that this distribution is purely determined by a dataset and can be readily calculated, and $P_\theta(\cdot)$ to indicate that this distribution is parameterized by $\theta$ and can be calculated readily (e.g., by forward propagation of a neural network). Calligraphic letters (e.g., $\mathcal{D}$) denote datasets.

We consider a multi-attribute dataset with training split $\mathcal{D}_{\text{train}}$ and test split $\mathcal{D}_{\text{test}}$, which can be seen as both drawn from the underlying data distribution $P_{\text{data}}(X, S, U)$, where $X$ is the high-dimensional original input data, $S = \{S_1, S_2, \ldots, S_M\}$ denotes a set of $M$ user-chosen private attributes annotated on $X$, and $U = \{U_1, U_2, \ldots, U_N\}$ denotes a set of $N$ user-chosen useful attributes annotated on $X$. For example, in the AudioMNIST dataset (Becker et al., 2018), $X$ can denote the high dimensional audio clips, and the user can choose $S = \{\text{"gender"}\}$ as a private attribute to remove, and choose $U = \{\text{"spoken digit"}\}$ as a useful attribute to preserve.

Following the assumptions made in Bertran et al. (2019), we assume that all attributes in $S, U$ follow finite categorical distributions, to ensure a finite mutual information between $X$ and each attribute. We also realistically assume that each attribute is deterministic when $X$ is given, namely $P(S_i|X)$ and $P(U_j|X)$ are degenerate distributions.

With above definitions and assumptions, we can now formally describe our goal in the information theory framework as to find the optimal data obfuscation algorithm, denoted as $P_\theta(X'|X)$, by solving the following optimization problem

$$\min_{P_\theta(X'|X)} L = \sum_{i=1}^{M} I(X'; S_i) - \lambda \sum_{j=1}^{N} I(X'; U_j) - \mu I(X'; X),$$
(1)

where the random variable $X'$ denotes the obfuscated data, and our obfuscation algorithm $P_\theta(X'|X)$ is parameterized by $\theta$. $I(\cdot, \cdot)$ denotes Shannon mutual information, and $\lambda$ and $\mu$ are two hyperparameters used to trade-off privacy protection and utility preservation. This optimization objective tries to simultaneously minimize the information leaked for $S_i$ in $X'$ to remove private attributes, maximize the information of $U_j$ in $X'$ to preserve useful attributes, and maximize the information of original data $X$ in $X'$ to preserve the general features of the data. To summarize the relationship of random variables $U, S, X$, and $X'$, we illustrate their probabilistic model in Figure 2. The information-theoretic optimization objective of Equation 1 is similar to the objectives used in Bertran et al. (2019); Malekzadeh et al. (2019); Chen et al. (2024). It also closely resembles the optimization objective in Privacy Funnel or Information Bottleneck literature with different focuses (Makhdoumi et al., 2014;

Tishby et al., 2000; Alemi et al., 2017; Hjelm et al., 2019).

## 3.2. The Vulnerability of Existing Private Attributes Protection Methods

State-of-the-art utility-preserving private attributes protection methods have demonstrated satisfactory performance in their respective papers in recent years. These methods are based on adversarial training, where the protector trains an adversarial classifier trying to correctly infer the private attribute $S_i$ from obfuscated data $X'$, and jointly trains the obfuscation algorithm $P_\theta(X'|X)$ to prevent the adversarial classifier from making correct inferences.

Nevertheless, as widely discussed in the neighboring fields of adversarial robustness (Carlini & Wagner, 2017; Athalye et al., 2018; Ilyas et al., 2019; Carlini et al., 2019), generated image/video detection (Yu et al., 2019; Wang et al., 2020; Masood et al., 2023), membership inference attacks (Carlini et al., 2022) and gradient inversion in federated learning (Geiping et al., 2020; Huang et al., 2021), adversarial training based methods share a common and critical weakness. That is, although their trained defender can safely defend against the jointly-trained adversary, they are vulnerable to potentially stronger or unseen adversaries, which can be obtained by certain tailored adaptive attacking strategies, or simply using longer training time, more computation power, larger datasets, etc.

Despite the existence of many advanced attacking strategies in these neighboring fields listed above, we empirically find out that in the context of private attributes protection, an even simpler and more realistic attacking method can effectively break the state-of-the-art methods, which is formally described below. Given an obfuscation algorithm $P_\theta(X'|X)$, the attacker may repetitively get an original sample and its associated attributes $(x, s, u)$ from $P_{\text{data}}(X, S, U)$; feeds the original sample $x$ into the obfuscation algorithm to get the corresponding obfuscated sample $x' \sim P_\theta(X'|X = x)$; and thus collect a dataset of $(x, s, u, x')$ tuples. Then, this collected dataset is utilized to train a new adversarial classifier in a supervised manner. In the rest of this paper, we call this attacking method the **Probing Attack**. The Probing Attack is realistic in that it does not have assumptions on the training protocol or model structures of the obfuscation algorithm, and can be applied to either deterministic or stochastic obfuscation algorithms.

We conduct a motivational experiment on the Motion Sense dataset to reveal Probing Attack's negative impact on existing methods, where we suppress "gender" and "ID" as private attributes and preserve "activity" as a useful attribute. We use the Normalized Accuracy Gain (NAG) proposed in Chen et al. (2024) as the metric, which is generally proportional to the accuracy of a classifier trained on $X'$ for each attribute. The obfuscation is considered better when

*Table 1.* Comparison of the NAG of baseline methods on Motion Sense. We suppress "gender" and "ID" as private attributes, while preserving "activity" as a useful attribute. NAG-Protector suggests that this NAG is calculated using the protector's adversarial classifier. NAG-Attacker suggests that this NAG is calculated using the attacker's adversarial classifier trained with Probing Attack.

| Method | NAG-Protector (%) | | | NAG-Attacker (%) | | |
|---|---|---|---|---|---|---|
| | gender ($\downarrow$) | ID ($\downarrow$) | activity ($\uparrow$) | gender ($\downarrow$) | ID ($\downarrow$) | activity ($\uparrow$) |
| ADV | 14.2±2.4 | 7.3±0.4 | 86.9±11.5 | 62.9±7.5 | 37.3±10.7 | 86.9±11.5 |
| GAP | 0.0±0.0 | 0.0±0.0 | 85.5±0.7 | 64.2±0.2 | 49.5±0.4 | 85.5±0.7 |
| MSDA | 0.0±0.1 | 5.9±0.9 | 93.0±0.5 | 65.4±1.6 | 46.6±1.5 | 93.0±0.5 |
| BDQ | 18.2±2.0 | 5.8±0.4 | 90.5±2.3 | 56.0±9.4 | 30.8±14.8 | 90.5±2.3 |
| PPDAR | 0.2±0.3 | 0.9±0.3 | 93.7±0.1 | 65.7±1.4 | 49.8±0.5 | 93.7±0.1 |
| MaSS | 1.3±0.6 | 1.2±0.2 | 93.8±0.1 | 65.1±0.5 | 49.8±0.2 | 93.8±0.1 |

NAG is lower for private attributes and when NAG is higher for useful attributes. For each private attribute, we calculate the NAG using both 1) the protector's adversarial classifier trained jointly with the obfuscation algorithm, and 2) the attacker's new adversarial classifier trained with a moderate Probing Attack setup after the obfuscation algorithm is deployed. The detailed descriptions of the NAG's definition and experimental settings can be found in Section 5.

We can observe from the results shown in Table 1 that, for private attributes, all methods achieved low NAG with the protector's adversarial classifier but significantly higher NAG with the attacker's adversarial classifier. These experiment results substantiate existing works' severe vulnerability to the simple Probing Attack method, indicating their impracticality in challenging real-world deployment.

To further examine adversarial training based works' vulnerability, we present an information-theoretic interpretation of this phenomenon in Appendix C.

## 4. Our Method: PASS

To build a data obfuscation model that is robust to Probing Attack, we propose **PASS**, Private Attributes protection with Stochastic data Substitution, which abandons adversarial training but adopts a novel idea of stochastic data substitution. Specifically, we propose first to draw a subset from the training dataset $\mathcal{D}_{\text{train}}$, which is denoted as the **substitution dataset** $\mathcal{D}_{\text{substitute}}$. Then, for each input original sample $x$, instead of transforming $x$ into an obfuscated sample, we propose to strategically replace $x$ with a sample $x'$ in the substitution dataset $\mathcal{D}_{\text{substitute}}$ according to our designed substitution probability, such that the attacker can not correctly infer the private attributes of the original sample $x$ from the substituted sample $x'$, but can still infer the useful attributes and some general features of $x$ from $x'$.

We propose to parameterize our substitution probability with a neural network, and then train it with our novel loss

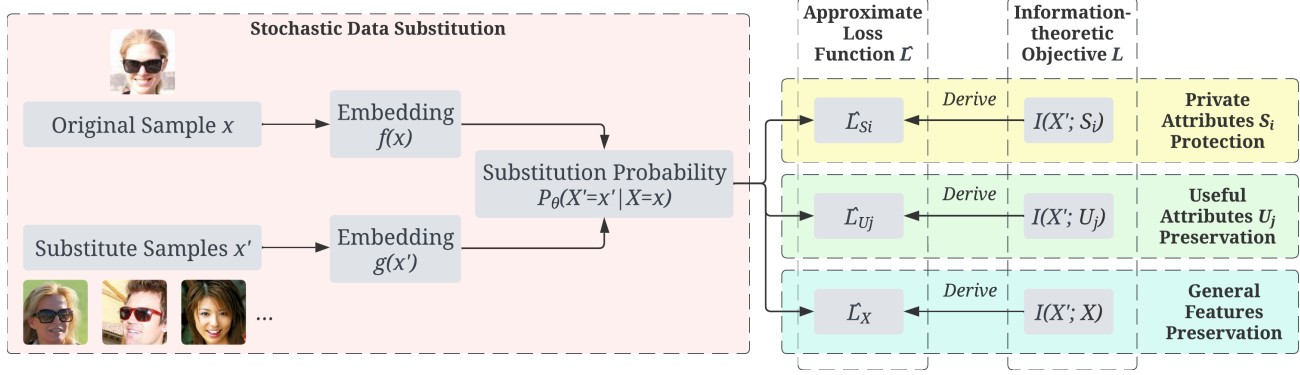

*Figure 3.* The overview of PASS design. PASS stochastically replaces each original sample $x$ with a substitute sample $x'$ according to probability $P_\theta(X' = x'|X = x)$. $P_\theta(X' = x'|X = x)$ is used to calculate approximate loss functions $\hat{L}_{S_i}$, $\hat{L}_{U_j}$ and $\hat{L}_X$, which are theoretically derived from $I(X'; S_i)$, $I(X'; U_j)$, and $I(X'; X)$ respectively, and are responsible for protecting private attribute $S_i$, preserving useful attribute $U_j$ and preserving general features, respectively.

function, which is rigorously derived from our information-theoretic objective Equation 1. An overview of PASS is shown in Figure 3, and the design details will be elaborated in the next sections.

### 4.1. Neural Network-based Stochastic Data Substitution

To unify the discussions of PASS and existing private attributes protection methods into the same information-theoretic framework, we reuse the notation $P_\theta(X'|X)$ to denote substitution probability, where $P_\theta(X' = x'|X = x)$ denotes the probability of substituting the original sample $x$ with the substitute sample $x' \in \mathcal{D}_{\text{substitute}}$, and $\theta$ denotes all the parameters in PASS.

To calculate $P_\theta(X'|X)$, we propose to first input each original input sample $x$ into a neural network to calculate an embedding, denoted as $f(x)$. Then, we obtain a learnable embedding $g(x')$ for each substitute sample $x' \in \mathcal{D}_{\text{substitute}}$. Next, we calculate $P_\theta(X'|X)$ based on the cosine similarity between each pair of embeddings $f(x)$ and $g(x')$ as

$$P_\theta(X' = x'|X = x) = \frac{e^{\cos(f(x), g(x'))/\tau}}{\sum_{x'' \in \mathcal{D}_{\text{substitute}}} e^{\cos(f(x), g(x''))/\tau}}, \quad (2)$$

where $\cos(\cdot, \cdot)$ is the cosine similarity, $\tau$ is the temperature hyper-parameter which is set to 0.01 for all of our experiments. Similar designs can be found widely in contrastive learning literature (Oord et al., 2018; Chen et al., 2020).

### 4.2. Loss Function Derivation

Ideally, to achieve our goal of utility-preserving private attributes protection, $P_\theta(X'|X)$ should be trained to minimize our information-theoretic optimization objective $L$ defined in Equation 1. Unfortunately, $L$ cannot be accurately esti-

mated for each mini-batch during training. Therefore, we propose a novel, fully differentiable loss function $\hat{L}$ to train $P_\theta(X'|X)$, which is derived step-by-step from $L$ and is theoretically valid in mini-batched based training.

In the rest of this section, we will focus on the formulation of $\hat{L}$. For a deeper understanding, please refer to Appendix D for detailed discussions on 1) the theoretical reason why $L$ cannot be accurately estimated, 2) the step-by-step derivation of $\hat{L}$ and proof of its theoretical validity, and 3) an intuitive explanation of $\hat{L}$ with a running example.

$\hat{L}$ can be decomposed into several loss terms as

$$\hat{L} = \sum_{i=1}^{M} \hat{L}_{S_i} - \lambda \sum_{j=1}^{N} \hat{L}_{U_j} - \mu\hat{L}_X, \quad (3)$$

where $\hat{L}_{S_i}$, $\hat{L}_{U_j}$ and $\hat{L}_X$ are derived from $I(X'; S_i)$, $I(X'; U_j)$, and $I(X'; X)$ respectively, and are responsible for protecting each private attribute $S_i$, preserving each useful attribute $U_j$ and preserving general features, respectively. $\lambda$ and $\mu$ are the same trade-off hyperparameters used in $L$. We will focus on each of them below.

**Private Attributes Protection.** To remove the information of each private attribute $S_i$ from $X'$, we minimize $I(X'; S_i)$ by minimizing $\hat{L}_{S_i}$, which is derived to be

$$\hat{L}_{S_i} = -\mathbb{E}_{P_{\text{data}}(S_i)}\left[H(X'|S_i)\right], \quad (4)$$

where $H(\cdot|\cdot)$ denotes conditional Shannon entropy. The conditional distribution $P(X'|S_i)$ is calculated as

$$P(X'|S_i) = \mathbb{E}_{P_{\text{data}}(X|S_i)}\left[P_\theta(X'|X)\right], \quad (5)$$

where $P_{\text{data}}(X|S_i)$ can be interpreted as all $x$ with each class of private attribute $S_i$ in the dataset. Taking expectation over $P_{\text{data}}(X|S_i)$ is estimated by averaging over all $x$ with $S_i$ in a mini-batch.

**Useful Attributes Preservation.** Extending the notations in Section 3.1, we denote the data distribution of $\mathcal{D}_{\text{substitute}}$ as $P_{\text{data}}(X', S', U')$, where $U' = \{U'_1, U'_2, \ldots, U'_N\}$ are called substitute useful attributes, which represents the useful attributes directly annotated on $X'$.

To preserve the useful attributes $U_j$, we propose to encourage the substitute useful attributes $U'_j$ to be similar to the original useful attribute $U_j$. For the AudioMNIST example where we choose "spoken digit" as the useful attribute, for original audio with the spoken digit "1", we try to encourage the substitute audio also to have the spoken digit "1". To achieve this, we propose to minimize $\hat{L}_{U_j}$ defined as:

$$\hat{L}_{U_j} = \log |\mathcal{U}_j| \, \mathbb{E}_{P_{\text{data}}(X,U_j)} \left[ -\log P(U'_j = U_j | X)) \right], \quad (6)$$

where $\mathcal{U}_j$ denotes the support of $U_j$, $|\cdot|$ denotes the cardinality. $\log |\mathcal{U}_j|$ is a coefficient to adjust $\hat{L}_{U_j}$, so that the scale of $\hat{L}_{U_j}$ matches $I(X'; U_j)$. Taking expectation over $P_{\text{data}}(X)$ is estimated by averaging over a mini-batch. $P(U'_j | X)$ denotes the expected $U'_j$ of $X$, which can be calculated as:

$$P(U'_j | X) = \mathbb{E}_{P_\theta(X'|X)} \left[ P_{\text{data}}(U'_j | X') \right]. \quad (7)$$

As shown in Appendix D.2, $\hat{L}_{U_j}$ can also be rigorously derived from $I(X'; U_j)$.

**General Feature Preservation.** To preserve general features, we maximize $I(X'; X)$ by minimizing the conditional entropy of $P_\theta(X'|X)$, which can be written as minimizing the loss term $\hat{L}_X$:

$$\hat{L}_X = \mathbb{E}_{P_{\text{data}}(X)} \left[ H(X'|X) \right], \quad (8)$$

where taking expectation over $P_{\text{data}}(X)$ is also estimated by averaging over a mini-batch.

Next, we will show that the expectation of $\hat{L}$ over mini-batches is proportional to an upperbound of $L$ defined in Equation 1, suggesting that minimizing $\hat{L}$ can lead to the minimization of $L$, confirming its theoretical validity.

**Theorem 4.1.** *For $L$ and $\hat{L}$ defined in Equation 1 and Equation 3 respectively, for $\mu \leq N$, we can have*

$$\mathbb{E}\left[\hat{L}\right] + C \geq L, \quad (9)$$

*where the expectation is taken over all mini-batches, $C$ is a constant defined as*

$$C = (M - \mu) \log(|\mathcal{D}_{substitute}|) - \lambda \sum_{j=1}^{N} H(U_j) + \lambda. \quad (10)$$

Please refer to Appendix D.2 for detailed proof. The constant $C$ is independent of model's parameters $\theta$, and can be calculated before training to estimate $L$ during training.

## 4.3. Training and Inference Procedures

Our design follows a standard two-phase machine learning workflow, consisting of training and inference. During training, we compute $P_\theta(X'|X)$ using Equation 2 for a mini-batch of samples from $\mathcal{D}_{\text{train}}$, then calculate our loss function according to Equation 3 and update the parameters $\theta$ via backpropagation. We summarize the pseudo-code for training in 1.

Once training is complete, inference proceeds by freezing the model parameters $\theta$. For each unseen test-set sample $x \in \mathcal{D}_{\text{test}}$, we compute the substitution probability $P_\theta(X' = x'|X = x)$ and draw a $x'$ accordingly to substitute $x$. The inference pseudo-code is detailed in Algorithm 2.

---

**Algorithm 1** PASS Training Pseudo-code

---

**Require:** training dataset $\mathcal{D}_{\text{train}}$, substitution dataset $\mathcal{D}_{\text{substitute}}$, model parameters $\theta$
**Ensure:** trained model parameters $\theta$
1: **while** not reached max number of epochs **do**
2:     sample a mini-batch from $\mathcal{D}_{\text{train}}$
3:     compute $f(x)$ for $x$ in mini-batch
4:     compute $g(x')$ for $x' \in \mathcal{D}_{\text{substitute}}$
5:     compute $P_\theta(X' = x'|X = x)$ using $f(x)$ and $g(x')$ (Equation 2)
6:     compute $\hat{L}$ using $P_\theta(X' = x'|X = x)$ (Equation 3)
7:     update $\theta$ with $\frac{\partial \hat{L}}{\partial \theta}$
8: **end while**
9: **return** $\theta$

---

**Algorithm 2** PASS Inference Pseudo-code

---

**Require:** original sample $x$, substitution dataset $\mathcal{D}_{\text{substitute}}$, model parameters $\theta$
**Ensure:** substituted sample $x'$
1: compute $f(x)$
2: compute $g(x')$ for $x' \in \mathcal{D}_{\text{substitute}}$
3: compute $P_\theta(X' = x'|X = x)$ using $f(x)$ and $g(x')$ (Equation 2)
4: draw $x' \in \mathcal{D}_{\text{substitute}}$ according to $P_\theta(X' = x'|X = x)$
5: **return** $x'$

---

PASS has lower training and inference computational overhead compared with state-of-the-art methods, because its $P_\theta(X'|X)$ is parameterized as an embedding extraction followed by a cosine similarity, whereas the state-of-the-art methods typically parameterize $P_\theta(X'|X)$ as an end-to-end data reconstruction neural network (Ronneberger et al., 2015; Cao et al., 2022), which may require more computation.

## 4.4. Theoretical Analysis for Entangled Attributes

Intuitively, when the useful attributes and private attributes are highly correlated or entangled, we need to sacrifice the utility of useful attributes to an extent to achieve private attributes protection. To theoretically describe this phenomenon in the information theory framework, and to provide an estimation of the extent of the utility sacrifice, we present the following theorem, revealing that both $I(X'; U_j)$ and $I(X'; X)$ are bounded by $I(X'; S_i)$ and cannot be arbitrarily large.

**Theorem 4.2.** *For any $i \in \{1, 2, \ldots, M\}$, any $\mathcal{J} \subseteq \{1, 2, \ldots, N\}$, and $\mathcal{U} = \{U_j | j \in \mathcal{J}\}$, we can have*

$$\sum_{j \in \mathcal{J}} I(X'; U_j) \leq I(X'; S_i) + H(\mathcal{U}|S_i) + C(\mathcal{U}), \quad (11)$$

*where $C(\cdot)$ denotes the total correlation, $H(\cdot|\cdot)$ denotes conditional Shannon entropy. We can also have*

$$I(X'; X) \leq I(X'; S_i) + H(X|S_i). \quad (12)$$

Please refer to Appendix A for detailed proof. This theorem can be viewed as an extension of the analysis in Chen et al. (2024). The terms $H(\mathcal{U}|S_i)$, $C(\mathcal{U})$, and $H(X|S_i)$ can all be calculated before training to estimate the extent of utility sacrifice, and to track the training progress.

## 4.5. Theoretical Interpretation within Local Differential Privacy Framework

As elaborated in Appendix B, under certain assumptions, our information-theoretic problem formulation can also be interpreted within the framework of local differential privacy (LDP) (Yang et al., 2023). In this context, PASS can be viewed as an LDP mechanism, more specifically, an extension of the classical randomized response methods. The distinctiveness of PASS from classical methods lies in its capability to operate over high-dimensional data domains, its explicit focus on utility preservation, and its foundation in information-theoretic principles.

## 5. Experiments

### 5.1. Experimental Setup

**Datasets.** We thoroughly evaluated PASS on three multi-attribute benchmark datasets, each representing a different application of a different modality. These datasets include AudioMNIST (Becker et al., 2018), containing recordings of human voices; Motion Sense (Malekzadeh et al., 2019), consisting of human activity sensory signals; and CelebA (Liu et al., 2015), containing facial images. More detailed descriptions of datasets can be found in Appendix E.1.

**Baselines.** We primarily compare PASS against six state-of-the-art baseline methods: ALR (Bertran et al., 2019),

GAP (Huang et al., 2018), MSDA (Malekzadeh et al., 2019), BDQ (Kumawat & Nagahara, 2022), PPDAR (Wu et al., 2020), and MaSS (Chen et al., 2024), whose detailed descriptions are presented in Section 2.

**Evaluation Metrics and Default Probing Attack's Setting.** To measure the performance of the obfuscation on imbalanced datasets, we adopt the metric Normalized Accuracy Gain (NAG) proposed by Chen et al. (2024), which can ensure that all attributes are measured on the same scale. The NAG for private attribute $S_i$ is defined as

$$\text{NAG}(S_i) = \max \left( 0, \frac{Acc(S_i) - Acc_{\text{guessing}}(S_i)}{Acc_{\text{no\_suppr.}}(S_i) - Acc_{\text{guessing}}(S_i)} \right), \quad (13)$$

where $Acc(S_i)$ is the accuracy of a classifier trained on the obfuscated data and the ground truth label of attribute $S_i$. $Acc_{\text{guessing}}(S_i)$ is the accuracy of the majority classifier for $S_i$, serving as a lower bound of the $Acc(S_i)$. And $Acc_{\text{no\_suppr.}}(S_i)$ is the accuracy of a classifier trained on the original data and the ground truth label of attribute $S_i$, serving as an upper bound of $Acc(S_i)$.

A critical difference between our evaluation protocol and baselines lies in that, we use the attacker's adversarial classifier trained with the Probing Attack to calculate NAG and thus measure the obfuscation performance, enhancing the practicality and realisticity of the evaluation. In the default Probing Attack's setting, the attacker has the API of the trained obfuscation model $P_\theta(X'|X)$ and its training dataset. The attacker adopts a medium-sized neural network based adversarial classifier and trains it on the obfuscated data using the Probing Attack.

To evaluate whether our method and baselines can preserve the data's general features, we hide some attributes during training and only reveal them during evaluation to verify if they can be preserved. We call these attributes **hidden useful attributes** and denote them as $F_k$ for $k \in \{1, \ldots, K\}$.

NAG for each useful attribute $\text{NAG}(U_j)$ and each hidden useful attribute $\text{NAG}(F_k)$ is defined and calculated in the same way as $\text{NAG}(S_i)$. Higher NAG suggests that this attribute's information is well preserved in $X'$. Therefore, the performance of an obfuscation method is considered better when each $\text{NAG}(S_i)$ is lower and when each $\text{NAG}(U_j)$ and each $\text{NAG}(F_k)$ is higher.

To facilitate a fair comparison between different methods, we propose a novel scalar metric, mean Normalized Accuracy Gain (mNAG), to measure the trade-off between private attributes protection and useful attributes preservation in a comprehensive way. mNAG is defined as the average of the NAG for useful attributes (including hidden useful attributes), minus the average of the NAG for private attributes,

*Table 2.* Comparison of the NAG between PASS and baselines on AudioMNIST. We suppress "gender" as a private attribute, while preserving "digit" as a useful attribute. We take "accent", "age", and "ID" as hidden useful attributes to evaluate general feature preservation.

| Method | NAG (%) | | | | | mNAG (%) (↑) |
|---|---|---|---|---|---|---|
| | gender (↓) | accent (↑) | age (↑) | ID (↑) | digit (↑) | |
| No suppr. | 100.0 | 100.0 | 100.0 | 100.0 | 100.0 | 0.0 |
| Guessing | 0.0 | 0.0 | 0.0 | 0.0 | 0.0 | 0.0 |
| ADV | 71.4±1.2 | 62.3±1.0 | 55.2±0.6 | 72.3±0.3 | 99.8±0.1 | 1.0±1.6 |
| GAP | 13.3±2.6 | 0.1±0.1 | 0.0±0.0 | 3.4±0.3 | 21.2±0.4 | -7.1±2.4 |
| MSDA | 78.4±2.9 | 61.9±3.3 | 57.3±3.0 | 77.1±2.4 | 99.8±0.0 | -4.3±0.8 |
| BDQ | 69.0±5.8 | 56.9±5.8 | 47.7±5.9 | 68.1±5.7 | 99.7±0.1 | -0.8±1.7 |
| PPDAR | 81.7±1.0 | 68.4±0.8 | 60.7±0.6 | 74.0±0.9 | 99.7±0.0 | -6.0±0.8 |
| MaSS | 88.9±1.2 | 76.0±0.7 | 70.4±1.0 | 81.1±0.3 | 99.5±0.1 | -7.2±0.8 |
| PASS | 0.0±0.0 | 46.4±0.9 | 27.6±1.6 | 49.7±0.4 | 96.5±0.2 | **55.0±0.7** |

*Table 3.* Comparison of the NAG of PASS on AudioMNIST for different configurations. In each configuration, the attributes annotated with (S) are suppressed as private attributes, and the attributes annotated with (U) are preserved as useful attributes.

| | NAG (%) | | | | | mNAG (%) (↑) |
|---|---|---|---|---|---|---|
| gender | accent | age | ID | digit | | |
| (S) 0.0±0.0 | (U) 65.2±0.4 | (U) 77.0±0.1 | (U) 73.3±0.4 | (U) 92.0±0.3 | | 76.9±0.2 |
| (S) 9.3±0.9 | (S) 0.0±0.0 | (U) 67.8±0.3 | (U) 61.9±0.2 | (U) 93.4±0.4 | | 69.7±0.4 |
| (S) 0.1±0.2 | (S) 0.0±0.0 | (S) 30.7±0.6 | (U) 40.0±0.4 | (U) 95.8±0.4 | | 57.6±0.2 |
| (S) 0.0±0.0 | (S) 0.0±0.0 | (S) 0.0±0.0 | (S) 0.0±0.0 | (U) 99.5±0.2 | | 99.5±0.2 |

which can be formally written as

$$
\text{mNAG} = \frac{1}{M+K}\left(\sum_{j=1}^{M}\text{NAG}(U_j) + \sum_{k=1}^{K}\text{NAG}(F_k)\right) \\
- \frac{1}{N}\sum_{i=1}^{N}\text{NAG}(S_i). \tag{14}
$$

We report the NAG and mNAG in the main paper, while the corresponding accuracy can be found in Appendix F.

**Hyperparameters.** Unless otherwise specified, we set $\lambda = \frac{N}{M}$ and $\mu = 0.2N$ throughout our experiments to balance private attributes protection, useful attributes preservation, and general feature preservation. The substitute dataset is constructed by randomly sampling 4096 data points from the training dataset. All the experiments in this paper are conducted with three random seeds and then aggregated. Other hyperparameters for neural network structures, training configurations, and datasets can be found in Appendix E.

### 5.2. Evaluation on Human Voice Recording

We begin with the task of removing gender information from human voice recordings on AudioMNIST dataset, where we suppress "gender", and preserve "digit". We take "accent", "age", "ID" as hidden useful attributes to evaluate general feature preservation. As shown in Table 2, PASS exhibited 0

*Table 4.* Comparison of the NAG between PASS and baselines on Motion Sense. We suppress "gender" and "ID" as private attributes, while preserving "activity" as a useful attribute. NAG-unfinetuned means that this NAG is calculated with a classifier that is only pre-trained on original data but not finetuned on substituted data.

| Method | NAG (%) | | | NAG-unfinetuned (%) | mNAG (%) (↑) |
|---|---|---|---|---|---|
| | gender (↓) | ID (↓) | activity (↑) | activity (↑) | |
| ADV | 62.9±7.5 | 37.3±10.7 | 86.9±11.5 | 93.8±0.7 | 36.8±4.3 |
| GAP | 64.2±0.2 | 49.5±0.4 | 85.5±0.7 | 10.2±3.5 | 28.6±0.6 |
| MSDA | 65.4±1.6 | 46.6±1.5 | 93.0±0.5 | 5.5±9.6 | 37.0±2.0 |
| BDQ | 56.0±9.4 | 30.8±14.8 | 90.5±2.3 | 0.0±0.0 | 47.1±9.3 |
| PPDAR | 65.7±1.4 | 49.8±0.5 | 93.7±0.1 | 0.0±0.0 | 36.0±0.8 |
| MaSS | 65.1±0.5 | 49.8±0.2 | 93.8±0.1 | 0.0±0.0 | 36.3±0.3 |
| PASS | 0.0±0.0 | 0.0±0.0 | 98.1±0.3 | 97.6±0.3 | **98.1±0.3** |

NAG on "gender" and a significantly higher mNAG than all baselines, which strongly substantiated PASS's capability on utility-preserving private attributes protection, and PASS's tolerance to the Probing Attack.

Furthermore, we conduct an ablation study to verify if PASS is robust to different combinations of useful and private attributes. The combinations and their corresponding results are presented in Table 3, demonstrating that PASS can consistently achieve a high mNAG for all combinations. When useful attributes and private attributes are highly entangled (e.g., when we suppress "gender", "accent","age", but preserve "ID"), PASS manages to find a satisfactory compromise between privacy and utility automatically.

We also conduct four more ablation studies on AudioMNIST by varying 1) the coefficient $\lambda$, 2) the coefficient $\mu$, 3) the number of samples in the substitute dataset $|\mathcal{D}_{\text{substitute}}|$, and 4) the distribution of the substitute dataset. As shown in Appendix F.1, PASS consistently maintains high performance across all different settings, showing robustness to hyperparameter changes.

### 5.3. Evaluation on Human Activity Sensory Data

In our subsequent study, we apply PASS to the task of anonymized activity recognition on Motion Sense dataset, where we suppress "gender" and "ID" attributes while preserving the "activity" attribute. On "activity", in addition to the standard NAG, we also report the NAG calculated with an un-finetuned classifier, which is only pre-trained on original data $X$, without finetuning on $X'$. The results, presented in Table 4, show that PASS achieves NAG of 0 for private attributes and a much higher mNAG than all baselines, which substantiated the effectiveness of PASS. Besides, PASS also achieved the highest NAG on "activity" with un-finetuned classifier, which shows that PASS can be plugged into an existing ML pipeline to protect private attributes without necessarily altering the downstream models. We visualize the results of data substitution in this experiment using

*Table 5.* Comparison of the NAG between PASS and baselines on CelebA. We suppress "Male" as a private attribute, while preserving "Smiling" and "Young" as useful attributes, and we take "Attractive," "Mouth_Slightly_Open," and "High_Cheekbones" as hidden useful attributes to evaluate general feature preservation.

| Method | NAG (%) | | | | | | mNAG (%) (↑) |
|--------|---------|---|---|---|---|---|--------------|
| | Male (↓) | Smiling (↑) | Young (↑) | Attractive (↑) | Mouth_Slightly_Open (↑) | High_Cheekbones (↑) | |
| ADV | 99.9±0.1 | 98.8±0.1 | 97.0±0.9 | 94.6±0.4 | 99.1±0.1 | 97.0±0.5 | -2.6±0.2 |
| GAP | 83.0±1.1 | 75.9±1.3 | 45.4±3.0 | 77.6±1.1 | 61.1±2.1 | 75.6±0.7 | -15.9±2.3 |
| MSDA | 91.6±0.7 | 99.8±0.2 | 92.4±2.4 | 89.9±1.0 | 91.8±0.8 | 95.7±1.1 | 2.3±0.8 |
| BDQ | 99.7±0.1 | 98.8±0.2 | 96.3±0.8 | 94.1±0.6 | 98.9±0.4 | 97.0±0.3 | -2.7±0.2 |
| PPDAR | 99.7±0.1 | 98.9±0.3 | 97.2±1.2 | 94.4±0.6 | 99.0±0.1 | 97.0±0.4 | -2.4±0.3 |
| MaSS | 96.9±0.1 | 97.2±0.2 | 86.2±1.4 | 90.6±0.3 | 97.6±0.2 | 94.6±0.4 | -3.7±0.4 |
| PASS | 4.9±0.5 | 98.3±0.1 | 78.6±0.8 | 58.1±2.8 | 67.0±0.8 | 86.7±0.3 | **72.9±0.2** |

confusion matrices, as shown in Appendix F.2.

We also experimented to show that PASS can safely tolerate the Probing Attack when the attacker has more data than the protector. Detailed results and analysis are presented in Appendix F.2.

### 5.4. Evaluation on Facial Images

We then extend the application of PASS to removing gender information from facial images on CelebA dataset, where we aim to suppress the "Male" as private attribute while preserving "Smiling", "Young" as useful attributes, and evaluating the general features preservation by taking "Attractive", "Mouth_Slightly_Open", "High_Cheekbones" as hidden useful attributes. As displayed in Table 5, PASS achieved a near 0 NAG on "Male", and a much higher mNAG than all the baselines, which highlights PASS's efficacy on images.

We further compare PASS with four additional DP-based baselines: Laplace Mechanism (Additive Noise) (Dwork et al., 2006), DPPix (Fan, 2018), Snow (John et al., 2020), and DP-Image (Xue et al., 2021). As shown in Table 17 in Appendix F.3, these methods show limited performance because they primarily aim to prevent membership inference, which differs from our goal of protecting specific private attributes while preserving utility.

In addition, we conduct an ablation study showing PASS's robustness on different combinations of useful and private attributes on CelebA. And we also experiment to reveal PASS's superiority over the baseline SPAct (Dave et al., 2022). Please refer to Appendix F.3 for results and analyses.

### 6. Conclusion

In this paper, we theoretically and empirically demonstrate that the common weakness of adversarial training has a noticeable negative impact on state-of-the-art private attributes protection methods. To address this, we propose PASS, a stochastic data substitution based method rooted rigorously in information theory, that overcomes the above weakness. The evaluation of PASS on three datasets substantiates PASS's broad applicability in various applications.

### Disclaimer

This paper was prepared for informational purposes with contributions from the Global Technology Applied Research center of JPMorgan Chase & Co. This paper is not a product of the Research Department of JPMorgan Chase & Co. or its affiliates. Neither JPMorgan Chase & Co. nor any of its affiliates makes any explicit or implied representation or warranty and none of them accept any liability in connection with this paper, including, without limitation, with respect to the completeness, accuracy, or reliability of the information contained herein and the potential legal, compliance, tax, or accounting effects thereof. This document is not intended as investment research or investment advice, or as a recommendation, offer, or solicitation for the purchase or sale of any security, financial instrument, financial product or service, or to be used in any way for evaluating the merits of participating in any transaction.

### Acknowledgment

Authors Yizhuo Chen and Tarek Abdelzaher were supported in part by the Boeing Company and in part by NSF CNS 20-38817.

### Impact Statement

We believe that there is no ethical concern or negative societal consequence related to this work. Our work benefits the protection of people's privacy in that it is proposed to remove the private attributes from the data in data sharing or processing pipelines.

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

# Appendix

## A. Proof of Theorem 4.2

*Proof.* For the first inequality of Theorem 4.2. For all $i \in \{1, \ldots, M\}$ and all $\mathcal{U} \subseteq \{U_1, \ldots, U_N\}$, we can have

$$
\begin{aligned}
&I(X'; S_i) + H(\mathcal{U}|S_i) + C(\mathcal{U}) \\
=&I(X'; \mathcal{U}, S_i) - I(X'; \mathcal{U}|S_i) + H(\mathcal{U}|S_i) + C(\mathcal{U}) \\
=&I(X'; \mathcal{U}, S_i) + H(\mathcal{U}|X', S_i) + C(\mathcal{U}) \\
=&I(X'; \mathcal{U}) + I(X'; S_i|\mathcal{U}) + H(\mathcal{U}|X', S_i) + C(\mathcal{U}) \\
\geq&I(X'; \mathcal{U}) + C(\mathcal{U}) \\
=&C(X', \mathcal{U}) \\
=&\sum_{U \in \mathcal{U}} I(X'; U) + C(\mathcal{U}|X') \\
\geq&\sum_{U \in \mathcal{U}} I(X'; U)
\end{aligned}
\tag{15}
$$

Similar to Chen et al. (2024), the second inequality of Theorem 4.2 can be proved as

$$
\begin{aligned}
I(X'; X) &= H(X') - H(X'|X) \\
&= H(X') - H(X'|X, S_i) \\
&\leq H(X') - H(X'|X, S_i) + H(X|X', S_i) \\
&= H(X') + H(X|S_i) - H(X'|S_i) \\
&= I(X'; S_i) + H(X|S_i)
\end{aligned}
\tag{16}
$$

$\square$

## B. Theoretical Analysis on the Connection with Local Differential Privacy

In this section, we will theoretically show that, under certain assumptions, our information-theoretic optimization objective in Equation 1 can be converted into a local differential privacy objective (Yang et al., 2023).

We first describe the definition of local differential privacy as follows. Let $\epsilon$ and $\delta$ be non-negative real numbers, and $\mathcal{A}$ be a randomized algorithm taking original sample $x$ from a dataset as input. Let $\text{im}\mathcal{A}$ denotes the image of $\mathcal{A}$. Then $\mathcal{A}$ satisfies $(\epsilon, \delta)$-local differential privacy if for any $\mathcal{S} \subseteq \text{im}\mathcal{A}$, any pair of original data points $x_1$ and $x_2$:

$$
P(\mathcal{A}(x_1) \in \mathcal{S}) \leq e^\epsilon P(\mathcal{A}(x_2) \in \mathcal{S}) + \delta
\tag{17}
$$

Next, we formulate our goal of private attributes suppression into the local differential privacy framework. We first assume an almighty attacker with the optimal adversarial classifier that can always infer each private attribute $S_i$ from substituted data $X'$ with the ground truth distribution $P(S_i|X')$. Then, for attribute $S_i$, we can define the randomized inference algorithm $\mathcal{A}(x)$ for each original sample $x$ as the almighty attacker's expected inference result on $x$:

$$
P(\mathcal{A}(x)) = \mathbb{E}_{P_\theta(X'|X=x)}[P(S_i|X')]
\tag{18}
$$

Our goal can then be expressed as training a substitution probability distribution $P_\theta(X'|X)$, such that $\mathcal{A}$ satisfies $(\epsilon, \delta)$-local differential privacy. This formulation means that, with our trained $P_\theta(X'|X)$, the almighty attacker's inference results on $S_i$ for any two data points $x_1$ and $x_2$ are similar, which means that even the almighty attacker cannot infer $S_i$ reliably for each $x$.

Finally, we show that our goal's information-theoretic formulation can be converted into our goal's local differential privacy formulation. Recall that we propose to minimize $I(X'; S_i)$ in Equation 1 to suppress $S_i$, which can be written as:

$$
I(X'; S_i) = \mathbb{E}_{P(X')}[KL(P(S_i|X')||P(S_i))]
\tag{19}
$$

Suppose that after training $P_\theta(X'|X)$ using this objective, we manage to achieve:

$$KL(P(S_i|X' = x')||P(S_i)) \leq \gamma \tag{20}$$

for any $x'$. Then we can use the following theorem to prove that our $\mathcal{A}(x)$ can achieve $(0, \sqrt{2\gamma})$-local differential privacy:

**Theorem B.1.** *For $\mathcal{A}(x)$ defined in Equation 18, if $KL(P(S_i|X' = x')||P(S_i)) \leq \gamma$ for any $x'$, then for any $\mathcal{S} \subseteq im\mathcal{A}$, and any $x_1$ and $x_2$ we can have*

$$P(\mathcal{A}(x_1) \in \mathcal{S}) \leq P(\mathcal{A}(x_2) \in \mathcal{S}) + \sqrt{2\gamma} \tag{21}$$

The proof of this theorem is as follows:

*Proof.* Using Pinsker's Inequality (Cover, 1999), for any $x'$, we can have

$$\sup_{\mathcal{S}} |P(S_i \in \mathcal{S}|X' = x') - P(S_i \in \mathcal{S})|$$
$$\leq \sqrt{\frac{1}{2}KL(P(S_i|X' = x')||P(S_i))} \tag{22}$$
$$\leq \sqrt{\frac{\gamma}{2}}$$

Therefore, for any $S$ and $x'$, we have:

$$P(S_i \in \mathcal{S}) - \sqrt{\frac{\gamma}{2}}$$
$$\leq P(S_i \in \mathcal{S}|X' = x') \tag{23}$$
$$\leq P(S_i \in \mathcal{S}) + \sqrt{\frac{\gamma}{2}}$$

Then, we can use this inequality to prove:

$$\begin{aligned}
P(\mathcal{A}(x_1) \in \mathcal{S}) &= \mathbb{E}_{P_\theta(X'|X=x_1)}\left[P(S_i \in \mathcal{S}|X')\right] \\
&\leq \mathbb{E}_{P_\theta(X'|X=x_1)}\left[P(S_i \in \mathcal{S}) + \sqrt{\frac{\gamma}{2}}\right] \\
&= P(S_i \in \mathcal{S}) + \sqrt{\frac{\gamma}{2}} \\
&= \mathbb{E}_{P_\theta(X'|X=x_2)}\left[P(S_i \in \mathcal{S}) + \sqrt{\frac{\gamma}{2}}\right] \\
&\leq \mathbb{E}_{P_\theta(X'|X=x_2)}\left[P(S_i \in \mathcal{S}|X') + \sqrt{2\gamma}\right] \\
&= P(\mathcal{A}(x_2) \in \mathcal{S}) + \sqrt{2\gamma}
\end{aligned} \tag{24}$$

$\square$

## C. Theoretical Analysis of Adversarial Training Based Methods' Vulnerability to The Probing Attack

In this section, we will theoretically uncover the underlying reason for adversarial training based methods' Vulnerability to The Probing Attack. Specifically, we first prove that the mutual information estimated and minimized by adversarial training based methods, denoted as $I_\phi(X'; S_i)$, is only a lower bound for the true mutual information $I(X'; S_i)$. The proof can be

written as:

$$
I(X'; S_i) - I_\phi(X'; S_i)
$$

$$
= \mathbb{E}_{P(X)P_\theta(X'|X)P(S_i|X)}[\log \frac{P(S_i|X')}{P_\phi(S_i|X')}]
$$

$$
= \mathbb{E}_{P_\theta(X')P(S_i|X')}[\log \frac{P(S_i|X')}{P_\phi(S_i|X')}] \tag{25}
$$

$$
= \mathbb{E}_{P_\theta(X')}[KL(P(S_i|X')||P_\phi(S_i|X'))]
$$

$$
\geq 0
$$

where $P_\phi(S_i|X')$ denotes the protector's adversarial classifier, parameterized by neural network $\phi$. Therefore, adversarial training based methods can not guarantee the minimization of $I(X'; S_i)$ by minimizing $I_\phi(X'; S_i)$. The remaining $I(X'; S_i)$ is unknown and potentially unbounded. If the attacker has a stronger adversarial classifier than $P_\phi(S_i|X')$, then the attacker can exploit the remaining $I(X'; S_i)$ to breach the protection.

## D. Detailed Explanations of Our Loss Function

### D.1. Theoretical Reason Why $L$ Cannot Be Accurately Estimated in Mini-batch

To understand why $L$ cannot be accurately estimated in mini-batch, we first introduce a random variable $B$, which denotes the index of the mini-batch. $B$ follows a uniform categorical distribution, where each index of mini-batch corresponds to a unique combination of samples in the mini-batch. We use $P(\cdot|B)$ to denote that this distribution is calculated only using the samples in mini-batch $B$.

During the training process, we can only calculate an estimation for mutual information terms in $L$ using all samples in each mini-batch. Taking $I(X'; S_i)$ as an example, let us denote the estimation in mini-batch $B$ as the conditional mutual information $I(X'; S_i|B)$. It can calculated as

$$
I(X'; S_i|B) = \mathbb{E}_{P(X',S_i|B)} \left[ \frac{P(X', S_i|B)}{P(X'|B)P(S_i|B)} \right] \tag{26}
$$

Then, let us denote the estimation of $L$ in each mini-batch $B$ as $L(B)$. It can be calculated as

$$
L(B) = \sum_{i=1}^{M} I(X'; S_i|B) - \lambda \sum_{j=1}^{N} I(X'; U_j|B) - \mu I(X'; X|B) \tag{27}
$$

Then, we can show that the expectation of $L(B)$ over all possible mini-batches is not equal to $L$

$$
\mathbb{E}_{P(B)}[L(B)]
$$

$$
= \mathbb{E}_{P(B)} \left[ \sum_{i=1}^{M} I(X'; S_i|B) - \lambda \sum_{j=1}^{N} I(X'; U_j|B) - \mu I(X'; X|B) \right]
$$

$$
\neq \sum_{i=1}^{M} I(X'; S_i) - \lambda \sum_{j=1}^{N} I(X'; U_j) - \mu I(X'; X) \tag{28}
$$

$$
= L
$$

where the inequality is achieved because the expectation of conditional mutual information is not necessarily equal to the mutual information.

In conclusion, mini-batched-based estimation $L(B)$ is not an unbiased estimator for $L$ and is not suitable for use as a loss function.

### D.2. Theoretical Derivation of Our Loss Function And Proof of of Theorem 4.1

In this section, we will keep using the random variable $B$ defined in Appendix D.1 to denote the index of mini-batch. We will rewrite our equations in Section 4.2 more formally with $B$ to elaborate our derivation and to prove Theorem 4.1. We

will omit the expectations when writing conditional entropy for clarity, except for $B$, whose expectations will still be written explicitly.

To derive $\hat{L}$ from $L$, we first need to derive the following upperbound for $L$

$$
\begin{aligned}
L &= \sum_{i=1}^{M} I(X'; S_i) - \lambda \sum_{j=1}^{N} I(X'; U_j) - \mu I(X'; X) \\
&= (M - \mu)H(X') - \sum_{i=1}^{M} H(X'|S_i) \\
&\quad - \lambda \sum_{j=1}^{N} H(U_j) + \lambda \sum_{j=1}^{N} H(U_j|X') + \mu H(X'|X) \\
&\leq (M - \mu)\log(|\mathcal{D}_{\text{substitute}}|) - \sum_{i=1}^{M} H(X'|S_i) \\
&\quad - \lambda \sum_{j=1}^{N} H(U_j) + \lambda \sum_{j=1}^{N} H(U_j|X') + \mu H(X'|X) \\
&= \sum_{i=1}^{M} -H(X'|S_i) + \lambda \sum_{j=1}^{N}(H(U_j|X') - 1) + \mu H(X'|X) \\
&\quad + C
\end{aligned}
\tag{29}
$$

where $H(\cdot)$ and $H(\cdot|\cdot)$ denote Shannon entropy and conditional Shannon entropy respectively, $C$ is a constant defined in Theorem 4.1. The inequality is achieved because the entropy of a random variable is smaller than or equal to the log cardinality of its support.

Next, we will focus on each term in Equation 29 separately.

**Private Attributes Protection**. In Equation 29, $-H(X'|S_i)$ is responsible for suppressing each private attribute $S_i$. Similar to the reasoning in Appendix D.1, we can calculate an estimation for $-H(X'|S_i)$ using all the samples in a mini-batch $B$. The estimation is denoted as $\hat{L}_{S_i}$, which can be rewritten with $B$ as

$$
\hat{L}_{S_i} = -H(X'|S_i, B)
\tag{30}
$$

The expectation of $\hat{L}_{S_i}$ over all mini-batches is an upperbound for $-H(X'|S_i)$

$$
\begin{aligned}
&- H(X'|S_i) \\
&\leq \mathbb{E}_{P(B)}[-H(X'|S_i, B)] \\
&= \mathbb{E}_{P(B)}[\hat{L}_{S_i}]
\end{aligned}
\tag{31}
$$

where the inequality is achieved because adding a condition cannot increase the entropy.

Therefore, we can manage to minimize $-H(X'|S_i)$ by calculating and then minimize $\hat{L}_{S_i}$.

**Useful Attributes Preservation**. In Equation 29, $H(U_j|X') - 1$ is responsible for preserving each useful attribute $U_j$. To show the connection between $H(U_j|X') - 1$ and $\hat{L}_{U_j}$, we need to first derive another upperbound for $H(U_j|X') - 1$, which can be written as

$$
\begin{aligned}
&H(U_j|X') - 1 \\
&= H(U_j|U_j', X') - 1 \\
&\leq H(U_j|U_j') - 1 \\
&\leq (1 - P(U_j' = U_j))\log|\mathcal{U}_j| + 1 - 1 \\
&= (1 - P(U_j' = U_j))\log|\mathcal{U}_j| \\
&\leq -\log P(U_j' = U_j)\log|\mathcal{U}_j| \\
&\leq \mathbb{E}_{P_{\text{data}}(X, U_j)}\left[-\log P(U_j' = U_j|X))\right]\log|\mathcal{U}_j|
\end{aligned}
\tag{32}
$$

where the first equation is achieved because $U'_j$ is independent of $U_j$ given $X'$. The first inequality is achieved because removing a condition cannot decrease the entropy. The second inequality is Fano's Inequality (Cover, 1999), and the fourth inequality is Jensen's Inequality.

When calculating an estimation of the above equation in a mini-batch $B$, we can have

$$P(U'_j = U_j|X, B) = P(U'_j = U_j|X) \tag{33}$$

which is because all the variables are independent of $B$ when $X$ is given. therefore, we can prove that $\hat{L}_{U_j}$ is an unbiased estimator as

$$
\begin{aligned}
&H(U_j|X') - 1 \\
\leq& \mathbb{E}_{P_{\text{data}}(X,U_j)} \left[ -\log P(U'_j = U_j|X)) \right] \log |\mathcal{U}_j| \\
=& \mathbb{E}_{P(B)} \mathbb{E}_{P_{\text{data}}(X,U_j)} \left[ -\log P(U'_j = U_j|X, B)) \right] \log |\mathcal{U}_j| \\
=& \mathbb{E}_{P(B)}[\hat{L}_{U_j}]
\end{aligned}
\tag{34}
$$

**General Features Preservation**. In Equation 29, $H(X'|X)$ is responsible for preserving general features. Similar to the analysis above, when calculating an estimation of $H(X'|X)$ in a mini-batch $B$, we can have

$$H(X'|X, B) = H(X'|X) \tag{35}$$

therefore, we can prove that $\hat{L}_X$ is an unbiased estimator as

$$
\begin{aligned}
&H(X'|X) \\
=& \mathbb{E}_{P(B)}[H(X'|X, B)] \\
=& \mathbb{E}_{P(B)}[\hat{L}_X]
\end{aligned}
\tag{36}
$$

In conclusion, summarizing the equations above, we can prove Theorem 4.1 as

$$
\begin{aligned}
&L \\
\leq& \sum_{i=1}^{M} -H(X'|S_i) + \lambda \sum_{j=1}^{N} (H(U_j|X') - 1) + \mu H(X'|X) + C \\
\leq& \mathbb{E}_{P(B)}[\hat{L}_{S_i}] + \lambda \mathbb{E}_{P(B)}[\hat{L}_{U_j}] + \mu \mathbb{E}_{P(B)}[\hat{L}_X] + C \\
=& \mathbb{E}_{P(B)}[\hat{L}] + C
\end{aligned}
\tag{37}
$$

### D.3. Intuitive Explanation of Our Loss Function with AudioMNIST Example

The intuition behind $\hat{L}_{S_i}$ is as follows. Maximizing the conditional entropy of $P(X'|S_i)$ may encourage that the original samples $x$ with each class of $S_i$ can be substituted with a wide range of $x' \in \mathcal{D}_{\text{substitute}}$, which may further ensure each $x'$ can substitute many different $x$ with different $S_i$ classes. Therefore, when the attacker observes a $x'$, it cannot confidently infer which $x$ with which class of $S_i$ is the original input sample, thus cannot infer the class of $S_i$ correctly.

Using the example of the AudioMNIST dataset, and supposing $S_i$ is "gender", our loss term $\hat{L}_{S_i}$ tries to encourage that each $x'$ can substitute both "male" speaker's audio and "female" speaker's audio, so that the attacker can not infer the speaker's gender of the $x$ when observing a $x'$.

To preserve the useful attributes $U_j$, we propose to encourage the substitute useful attributes $U'_j$ to be similar to the original useful attribute $U_j$. For the AudioMNIST example, supposing we choose "spoken digit" as the useful attribute, if there comes an audio with the spoken digit "1", then we try to encourage the substitute audio also to have the spoken digit "1".

$\hat{L}_X$ may encourage each original sample $x$ to be substituted by a narrow range of $x' \in \mathcal{D}_{\text{substitute}}$, which has a counteracting effect on the loss term $\hat{L}_{S_i}$. When $\hat{L}_X$ and $\hat{L}_{S_i}$ are both used to train $P_\theta(X'|X)$, their combined effect is to encourage that, although each $x$ can only cover a relatively narrow range of $x'$, all the $x$ with each class of $S_i$ may jointly cover a wide range of $x'$. Consequently, each $x'$ may only substitute a narrow range of $x$, but these $x$ are with different classes of $S_i$,

*Table 6.* Detailed configurations of our experiments' datasets, models, and optimization techniques.

| Experiment | Audio | Human activity | Facial image |
|---|---|---|---|
| Dataset | AudioMNIST | Motion Sense | CelebA |
| # total data points | 30000 | 74324 | 202599 |
| Training-testing split | 4:1 | 7:4 | 4:1 |
| Optimizer | AdamW (Loshchilov & Hutter, 2019) | | |
| Learning rate | 0.001 | 0.001 | 0.0001 |
| Weight decay | 0.0001 | | |
| Learning rate scheduler | Cosine scheduler | | |
| Embeddings $f(x)$ and $g(x')$ dimension | 512 | | |
| $P_\theta(X'\|X)$ training epochs | 2000 | 200 | 50 |
| Probing Attack training epochs | 2000 | 200 | 50 |
| $f(x)$ neural network structure | 3-layer MLP | 6-layer Convolutional NN | Pre-trained FaceNet backbone followed by 2-layer MLP |

which still hinders the attacker from inferring $S_i$ from $x'$, while ensuring that the downstream user can infer $x$ from $x'$ with medium level of accuracy.

Keep using the AudioMNIST dataset with the private attribute "gender" as an example. With both $\hat{L}_X$ and $\hat{L}_{S_i}$, we encourage that each $x'$ can only substitute a limited number of $x$, but these $x$ contain both "male" speaker's audio and "female" speaker's audio.

# E. Additional Experimental Setup

## E.1. Datasets Desciptions

**CelebA Dataset** comprises 202,599 facial images, each annotated with 40 binary attributes. For our experiments, we selected six representative attributes. We used the official split for training and validation. All images were center-cropped to a resolution of 160×160 pixels in preprocessing.

**AudioMNIST Dataset** includes human voice recordings in English, which contains 60 speakers speaking 10 digits. It is annotated with eight attributes. We selected attributes gender, accent, age, ID, and spoken digits for our experiments, which have 2, 16, 18, 60, and 10 classes, respectively. The dataset contains 30,000 audio clips, divided into 24,000 for training and 6,000 for validation. We enhance the experiment efficiency on the AudioMNIST dataset by converting the raw data into features using HuBERT-B (Hsu et al., 2021).

**Motion Sense Dataset** consists of accelerometer and gyroscope data recorded during six daily human activities. We focused on three attributes: gender, ID, and activity, with 2, 24, and 6 classes, respectively. Following Malekzadeh et al. (2019), we excluded the "sit" and "stand up" activities from our experiments; adopted the "trial" split strategy; used only the magnitudes of the gyroscope and accelerometer as input; normalized input to zero mean and unit standard deviation; and segmented the datasets into 74,324 samples, each with a length of 128.

## E.2. Model and Optimization Configurations

To achieve better training stability and faster convergence, we adopt a pre-trained FaceNet (Schroff et al., 2015) backbone followed by 2-layer MLPs as the neural network for CelebA dataset. We used a 6-layer Convolutional neural network for the Motion Sense dataset. Please refer to Table 6 for more detailed configurations of our experiments' datasets, models, and optimization techniques.

*Table 7.* Comparison of the NAG and accuracy between PASS and baselines on AudioMNIST. We suppress "gender" as a private attribute, while preserving "digit" as a useful attribute. We take "accent", "age", and "ID" as hidden useful attributes to evaluate general feature preservation.

| Method | NAG (Accuracy) (%) | | | | | mNAG (%) ($\uparrow$) |
|---|---|---|---|---|---|---|
| | gender ($\downarrow$) | accent ($\uparrow$) | age ($\uparrow$) | ID ($\uparrow$) | digit ($\uparrow$) | |
| No suppr. | 100.0 (99.7) | 100.0 (98.6) | 100.0 (97.1) | 100.0 (98.6) | 100.0 (99.9) | 0.0 |
| Guessing | 0.0 (80.0) | 0.0 (68.3) | 0.0 (16.7) | 0.0 (1.7) | 0.0 (10.0) | 0.0 |
| ADV | 71.4±1.2 (94.1±0.2) | 62.3±1.0 (66.8±0.8) | 55.2±0.6 (85.1±0.2) | 72.3±0.3 (71.8±0.3) | 99.8±0.1 (99.6±0.1) | 1.0±1.6 |
| GAP | 13.3±2.6 (82.6±0.5) | 0.1±0.1 (16.7±0.1) | 0.0±0.0 (68.3±0.0) | 3.4±0.3 (4.9±0.3) | 21.2±0.4 (29.0±0.3) | -7.1±2.4 |
| MSDA | 78.4±2.9 (95.5±0.6) | 61.9±3.3 (66.5±2.6) | 57.3±3.0 (85.7±0.9) | 77.1±2.4 (76.4±2.3) | 99.8±0.0 (99.6±0.0) | -4.3±0.8 |
| BDQ | 69.0±5.8 (93.6±1.1) | 56.9±5.8 (62.4±4.7) | 47.7±5.9 (82.8±1.8) | 68.1±5.7 (67.7±5.5) | 99.7±0.1 (99.6±0.1) | -0.8±1.7 |
| PPDAR | 81.7±1.0 (96.1±0.2) | 68.4±0.8 (71.7±0.7) | 60.7±0.6 (86.7±0.2) | 74.0±0.9 (73.4±0.9) | 99.7±0.0 (99.6±0.0) | -6.0±0.8 |
| MaSS | 88.9±1.2 (97.5±0.2) | 76.0±0.7 (77.8±0.5) | 70.4±1.0 (89.6±0.3) | 81.1±0.3 (80.3±0.3) | 99.5±0.1 (99.4±0.1) | -7.2±0.8 |
| PASS | 0.0±0.0 (79.9±0.1) | 46.4±0.9 (54.0±0.7) | 27.6±1.6 (76.7±0.5) | 49.7±0.4 (49.8±0.4) | 96.5±0.2 (96.7±0.2) | **55.0±0.7** |

*Table 8.* Comparison of the NAG and accuracy of PASS on AudioMNIST for different configurations. In each configuration, the attributes annotated with (S) are suppressed as private attributes, and the attributes annotated with (U) are preserved as useful attributes.

| Method | NAG (Accuracy) (%) | | | | | mNAG (%) ($\uparrow$) |
|---|---|---|---|---|---|---|
| | gender | accent | age | ID | digit | |
| No suppr. | (U) 100.0 (99.7) | (U) 100.0 (98.6) | (U) 100.0 (97.1) | (U) 100.0 (98.6) | (U) 100.0 (99.9) | 0.0 |
| Guessing | (S) 0.0 (80.0) | (S) 0.0 (68.3) | (S) 0.0 (16.7) | (S) 0.0 (1.7) | (S) 0.0 (10.0) | 0.0 |
| PASS | (S) 0.0±0.0 (79.4±0.2) | (U) 65.2±0.4 (88.1±0.1) | (U) 77.0±0.1 (78.6±0.1) | (U) 73.3±0.4 (72.7±0.4) | (U) 92.0±0.3 (92.7±0.2) | 76.9±0.2 |
| | (S) 9.3±0.9 (81.8±0.2) | (S) 0.0±0.0 (67.9±0.1) | (U) 61.8±0.3 (71.2±0.2) | (U) 61.9±0.2 (61.6±0.2) | (U) 93.4±0.4 (93.9±0.4) | 69.7±0.4 |
| | (S) 0.1±0.2 (79.6±0.4) | (S) 0.0±0.0 (67.4±0.4) | (S) 30.7±0.6 (41.4±0.5) | (U) 40.0±0.4 (40.4±0.4) | (U) 95.8±0.4 (96.1±0.4) | 57.6±0.2 |
| | (S) 0.0±0.0 (80.0±0.0) | (S) 0.0±0.0 (68.3±0.0) | (S) 0.0±0.0 (16.7±0.0) | (S) 0.0±0.0 (1.7±0.0) | (U) 99.5±0.2 (99.4±0.2) | 99.5±0.2 |

*Table 9.* Ablation study of varying the coefficient $\lambda$ on AudioMNIST. We suppress "gender" as a private attribute, while preserving "digit" as a useful attribute. We take "accent", "age", and "ID" as hidden useful attributes to evaluate general feature preservation.

| Method | $\lambda$ | NAG (Accuracy) (%) | | | | | mNAG (%) ($\uparrow$) |
|---|---|---|---|---|---|---|---|
| | | gender ($\downarrow$) | accent ($\uparrow$) | age ($\uparrow$) | ID ($\uparrow$) | digit ($\uparrow$) | |
| No suppr. | - | 100.0 (99.7) | 100.0 (98.6) | 100.0 (97.1) | 100.0 (98.6) | 100.0 (99.9) | 0.0 |
| Guessing | - | 0.0 (80.0) | 0.0 (68.3) | 0.0 (16.7) | 0.0 (1.7) | 0.0 (10.0) | 0.0 |
| PASS | 0.1 $N/M$ | 0.0±0.0 (80.0±0.0) | 45.0±0.5 (52.9±0.4) | 25.2±1.5 (76.0±0.4) | 48.5±0.6 (48.7±0.6) | 88.5±0.7 (89.5±0.6) | 51.8±0.7 |
| | 0.2 $N/M$ | 0.0±0.0 (80.0±0.0) | 45.3±0.3 (53.1±0.2) | 25.7±1.4 (76.1±0.4) | 49.3±0.1 (49.4±0.1) | 91.5±0.6 (92.2±0.5) | 53.0±0.4 |
| | 0.5 $N/M$ | 0.0±0.0 (80.0±0.0) | 47.3±0.7 (54.7±0.6) | 28.4±1.1 (76.9±0.3) | 50.8±0.5 (50.9±0.5) | 95.0±0.2 (95.4±0.2) | 55.4±0.6 |
| | 1 $N/M$ | 0.0±0.0 (79.9±0.1) | 46.4±0.9 (54.0±0.7) | 27.6±1.6 (76.7±0.5) | 49.7±0.4 (49.8±0.4) | 96.5±0.2 (96.7±0.2) | 55.0±0.7 |
| | 2 $N/M$ | 0.0±0.0 (80.0±0.0) | 47.8±0.5 (55.1±0.4) | 28.5±0.4 (77.0±0.1) | 51.3±0.7 (51.4±0.7) | 97.8±0.0 (97.9±0.0) | 56.4±0.4 |
| | 5 $N/M$ | 0.1±0.1 (80.0±0.0) | 46.2±0.7 (53.8±0.5) | 27.0±1.4 (76.5±0.4) | 49.9±0.8 (50.0±0.8) | 99.0±0.1 (98.9±0.1) | 55.4±0.5 |

*Table 10.* Ablation study of varying the coefficient $\mu$ on AudioMNIST. We suppress "gender" as a private attribute, while preserving "digit" as a useful attribute. We take "accent", "age", and "ID" as hidden useful attributes to evaluate general feature preservation.

| Method | $\mu$ | NAG (Accuracy) (%) | | | | | mNAG (%) ($\uparrow$) |
|---|---|---|---|---|---|---|---|
| | | gender ($\downarrow$) | accent ($\uparrow$) | age ($\uparrow$) | ID ($\uparrow$) | digit ($\uparrow$) | |
| No suppr. | - | 100.0 (99.7) | 100.0 (98.6) | 100.0 (97.1) | 100.0 (98.6) | 100.0 (99.9) | 0.0 |
| Guessing | - | 0.0 (80.0) | 0.0 (68.3) | 0.0 (16.7) | 0.0 (1.7) | 0.0 (10.0) | 0.0 |
| PASS | 0.00 | 0.0±0.0 (80.0±0.0) | 0.0±0.0 (16.7±0.0) | 0.0±0.0 (68.3±0.0) | 0.0±0.0 (1.7±0.0) | 97.3±0.4 (97.4±0.4) | 24.3±0.1 |
| | 0.01 | 0.0±0.0 (80.0±0.0) | 2.9±0.3 (19.0±0.2) | 0.0±0.0 (68.3±0.0) | 6.4±0.1 (7.9±0.1) | 99.8±0.1 (99.7±0.1) | 27.3±0.1 |
| | 0.02 | 0.0±0.0 (80.0±0.0) | 7.5±0.8 (22.7±0.6) | 0.0±0.0 (68.3±0.0) | 12.7±0.1 (13.9±0.1) | 99.8±0.0 (99.6±0.0) | 30.0±0.2 |
| | 0.05 | 0.0±0.0 (80.0±0.0) | 23.5±0.5 (35.6±0.4) | 4.1±0.3 (69.6±0.1) | 28.5±0.3 (29.2±0.3) | 98.9±0.2 (98.9±0.1) | 38.8±0.3 |
| | 0.10 | 0.0±0.0 (80.0±0.0) | 40.5±0.7 (49.3±0.6) | 20.4±1.8 (74.5±0.5) | 44.3±0.7 (44.6±0.7) | 97.9±0.1 (97.9±0.0) | 50.8±0.6 |
| | 0.20 | 0.0±0.0 (79.9±0.1) | 46.4±0.9 (54.0±0.7) | 27.6±1.6 (76.7±0.5) | 49.7±0.4 (49.8±0.4) | 96.5±0.2 (96.7±0.2) | **55.0±0.7** |
| | 0.50 | 0.0±0.0 (79.8±0.1) | 44.9±0.2 (52.8±0.1) | 25.1±1.3 (75.9±0.4) | 48.5±0.5 (48.6±0.5) | 91.4±0.9 (92.1±0.8) | 52.4±0.6 |

*Table 11.* Ablation study of varying the number of samples in the substitute dataset (denoted as $|\mathcal{D}_{\text{substitute}}|$) on AudioMNIST. We suppress "gender" as a private attribute, while preserving "digit" as a useful attribute. We take "accent", "age", and "ID" as hidden useful attributes to evaluate general feature preservation.

| Method | $|\mathcal{D}_{\text{substitute}}|$ | NAG (Accuracy) (%) | | | | | mNAG (%) (↑) |
|---|---|---|---|---|---|---|---|
| | | gender (↓) | accent (↑) | age (↑) | ID (↑) | digit (↑) | |
| No suppr. | - | 100.0 (99.7) | 100.0 (98.6) | 100.0 (97.1) | 100.0 (98.6) | 100.0 (99.9) | 0.0 |
| Guessing | - | 0.0 (80.0) | 0.0 (68.3) | 0.0 (16.7) | 0.0 (1.7) | 0.0 (10.0) | 0.0 |
| PASS | 1024 | 0.0±0.0 (80.0±0.0) | 41.6±0.4 (50.1±0.4) | 20.6±0.5 (74.6±0.1) | 45.6±0.3 (45.9±0.3) | 96.8±0.4 (97.0±0.4) | 51.1±0.2 |
| | 2048 | 0.0±0.0 (80.0±0.0) | 45.8±0.9 (53.5±0.7) | 24.9±1.6 (75.9±0.5) | 49.0±0.7 (49.2±0.7) | 96.6±0.4 (96.8±0.4) | 54.1±0.7 |
| | 4096 | 0.0±0.0 (79.9±0.1) | 46.4±0.9 (54.0±0.7) | 27.6±1.6 (76.7±0.5) | 49.7±0.4 (49.8±0.4) | 96.5±0.2 (96.7±0.2) | **55.0±0.7** |
| | 8192 | 0.0±0.1 (80.0±0.0) | 46.1±0.4 (53.7±0.3) | 26.1±2.0 (76.2±0.6) | 49.8±0.2 (50.0±0.2) | 96.7±0.1 (96.9±0.1) | 54.6±0.6 |
| | 16384 | 0.0±0.0 (80.0±0.0) | 46.2±1.1 (53.8±0.9) | 25.5±1.2 (76.0±0.4) | 49.5±0.6 (49.7±0.5) | 96.8±0.3 (97.0±0.2) | 54.5±0.2 |
| | 24000 | 0.0±0.0 (80.0±0.0) | 45.7±0.7 (53.4±0.6) | 26.7±0.6 (76.4±0.2) | 49.1±0.8 (49.2±0.7) | 97.1±0.2 (97.2±0.2) | 54.6±0.4 |

*Table 12.* Ablation study of varying the attribute distribution of "gender" and "digit" on substitute dataset on AudioMNIST. We suppress "gender" as a private attribute, while preserving "digit" as a useful attribute. We take "accent", "age", and "ID" as hidden useful attributes to evaluate general feature preservation.

| Method | "gender" distribution | "digit" distribution | NAG (Accuracy) (%) | | | | | mNAG (%) (↑) |
|---|---|---|---|---|---|---|---|---|
| | | | gender (↓) | accent (↑) | age (↑) | ID (↑) | digit (↑) | |
| No suppr. | - | - | 100.0 (99.7) | 100.0 (98.6) | 100.0 (97.1) | 100.0 (98.6) | 100.0 (99.9) | 0.0 |
| Guessing | - | - | 0.0 (80.0) | 0.0 (68.3) | 0.0 (16.7) | 0.0 (1.7) | 0.0 (10.0) | 0.0 |
| PASS | 90% Male 10% Female | 10% 0-9 | 0.0±0.0 (80.0±0.0) | 47.9±0.5 (55.2±0.4) | 28.4±0.7 (76.9±0.2) | 51.2±0.2 (51.3±0.2) | 96.9±0.3 (97.0±0.3) | 56.1±0.2 |
| | 80% Male 20% Female | 10% 0-9 | 0.0±0.0 (79.9±0.1) | 46.4±0.9 (54.0±0.7) | 27.6±1.6 (76.7±0.5) | 49.7±0.4 (49.8±0.4) | 96.5±0.2 (96.7±0.2) | 55.0±0.7 |
| | 50% Male 50% Female | 10% 0-9 | 0.1±0.2 (80.0±0.1) | 47.0±0.4 (54.5±0.3) | 28.1±0.7 (76.8±0.2) | 50.5±0.5 (50.6±0.5) | 96.5±0.3 (96.7±0.3) | 55.4±0.1 |
| | 20% Male 80% Female | 10% 0-9 | 0.0±0.1 (80.0±0.0) | 47.4±0.9 (54.8±0.8) | 27.8±3.4 (76.7±1.0) | 50.8±0.8 (50.9±0.8) | 96.7±0.2 (96.8±0.2) | 55.6±1.2 |
| | 10% Male 90% Female | 10% 0-9 | 0.0±0.1 (80.0±0.0) | 47.8±0.6 (55.1±0.5) | 27.4±0.7 (76.6±0.2) | 51.2±0.2 (51.3±0.2) | 96.3±0.1 (96.6±0.1) | 55.7±0.1 |
| | 80% Male 20% Female | 30% 0 7.8% 1-9 | 0.0±0.0 (79.9±0.0) | 46.7±0.5 (54.2±0.4) | 27.6±0.5 (76.7±0.1) | 50.0±0.5 (50.1±0.5) | 96.0±0.3 (96.3±0.3) | 55.1±0.1 |
| | 80% Male 20% Female | 50% 0 5.6% 1-9 | 0.0±0.0 (79.9±0.0) | 44.3±0.2 (52.3±0.1) | 25.3±0.5 (76.0±0.2) | 48.1±0.3 (48.3±0.3) | 93.9±0.4 (94.4±0.4) | 52.9±0.1 |

# F. Additional Experiment results

### F.1. Additional Results on AudioMNIST

In addition to measuring the experiment results in NAG in Table 2 and Table 3, we also measure these experiment results in accuracy, as shown in Table 7 and Table 8.

We then conduct ablation studies to show PASS's hyper-parameters stability. Our first ablation study adopts the same setting as in Table 7, except that we gradually change $\lambda$ from $0.1N/M$ to $5N/M$ to examine its impact on PASS's performance. The results in Table 9 show that PASS consistently achieves near-0 NAG on "gender" and stable overall performance across a wide range of values.

Similarly, our second ablation study varies $\mu$ from 0 to 0.50 to evaluate its influence on PASS. As shown in Table 10, PASS consistently achieves 0 NAG on the private attribute "gender" for all $\mu$. The NAG for the useful attribute "digit" peaks at $\mu = 0.01$, while the NAG for other hidden useful attributes peaks at a much higher value of $\mu = 0.20$. This behavior is well expected because $\mu$ is designed as the coefficient for $I(X'; X)$, where higher $\mu$ tends to trade useful attributes preservation for general feature preservation.

Next, our third ablation study evaluates PASS's stability to the number of samples in the substitution dataset. Again, we adopt the setting as Table 7, except that we gradually change the number of samples in the substitution dataset from 1024 to 24000 (the training dataset has 24000 samples). As shown in Table 11, PASS obtains highly consistent mNAG for all different numbers of samples.

Finally, in our fourth ablation study, we construct multiple substitution datasets with varying attribute distributions: 1) for the sensitive attribute "gender", we varied its distribution from "90% Male / 10% Female" to "10% Male / 90% Female"; 2) for the useful attribute "digit", we varied its distribution from uniform distribution "10% 0-9" to a highly skewed distribution "50% 0 / 5.6% 1-9". The results, shown in Table A below, demonstrate that PASS consistently maintains high performance across all these different substitution dataset configurations, even under highly imbalanced conditions.

These four ablation studies provide empirical evidence that PASS has strong stability against the variance of this hyper-parameter.

## F.2. Additional Results on Motion Sense

*Table 13.* Comparison of the NAG and accuracy of baseline methods on Motion Sense. We suppress "gender" and "ID" as private attributes, while preserving "activity" as a useful attribute. NAG-Protector suggests that this NAG is calculated using the protector's adversarial classifier. NAG-Attacker suggests that this NAG is calculated using the attacker's adversarial classifier trained with Probing Attack.

| Method | NAG (Accuracy) (%) | | | | |
| | gender (↓) | | ID (↓) | | activity (↑) |
| | protector's classifier | attacker's classifier | protector's classifier | attacker's classifier | |
|---|---|---|---|---|---|
| ADV | 14.2±2.4 (62.8±1.0) | 62.9±7.5 (82.8±3.1) | 7.3±0.4 (11.7±0.3) | 37.3±10.7 (37.5±9.2) | 86.9±11.5 (91.0±5.9) |
| GAP | 0.0±0.0 (57.0±0.0) | 64.2±0.2 (83.3±0.1) | 0.0±0.0 (4.7±0.1) | 49.5±0.4 (48.1±0.3) | 85.5±0.7 (90.3±0.3) |
| MSDA | 0.0±0.1 (57.0±0.1) | 65.4±1.6 (83.8±0.7) | 5.9±0.9 (10.5±0.8) | 46.6±1.5 (45.6±1.3) | 93.0±0.5 (94.2±0.3) |
| BDQ | 18.2±2.0 (64.5±0.8) | 56.0±9.4 (79.9±3.9) | 5.8±0.4 (10.3±0.4) | 30.8±14.8 (31.9±12.8) | 90.5±2.3 (92.9±1.1) |
| PPDAR | 0.2±0.3 (56.9±0.4) | 65.7±1.4 (83.9±0.6) | 0.9±0.3 (6.1±0.2) | 49.8±0.5 (48.3±0.4) | 93.7±0.1 (94.5±0.1) |
| MaSS | 1.3±0.6 (57.5±0.3) | 65.1±0.5 (83.7±0.2) | 1.2±0.2 (6.4±0.1) | 49.8±0.2 (48.4±0.2) | 93.8±0.1 (94.6±0.0) |

*Table 14.* Comparison of the NAG and accuracy between PASS and baselines on Motion Sense. We suppress "gender" and "ID" as private attributes, while preserving "activity" as useful attributes. NAG-unfinetuned means that this NAG is calculated with a classifier that is only pre-trained on original data but not finetuned on substituted data.

| Method | NAG (Accuracy) (%) | | | NAG-unfinetuned (Accuracy) (%) | mNAG (%) (↑) |
| | gender (↓) | ID (↓) | activity (↑) | activity (↑) | |
|---|---|---|---|---|---|
| No suppr. | 100.0 (98.0) | 100.0 (91.7) | 100.0 (97.8) | 100.0 (97.8) | 0.0 |
| Guessing | 0.0 (57.0) | 0.0 (5.3) | 0.0 (46.6) | 0.0 (46.6) | 0.0 |
| ADV | 62.9±7.5 (82.8±3.1) | 37.3±10.7 (37.5±9.2) | 86.9±11.5 (91.0±5.9) | 93.8±0.7 (94.6±0.4) | 36.8±4.3 |
| GAP | 64.2±0.2 (83.3±0.1) | 49.5±0.4 (48.1±0.3) | 85.5±0.7 (90.3±0.3) | 10.2±3.5 (51.8±1.8) | 28.6±0.6 |
| MSDA | 65.4±1.6 (83.8±0.7) | 46.6±1.5 (45.6±1.3) | 93.0±0.5 (94.2±0.3) | 5.5±9.6 (42.5±13.0) | 37.0±2.0 |
| BDQ | 56.0±9.4 (79.9±3.9) | 30.8±14.8 (31.9±12.8) | 90.5±2.3 (92.9±1.1) | 0.0±0.0 (22.6±9.2) | 47.1±9.3 |
| PPDAR | 65.7±1.4 (83.9±0.6) | 49.8±0.5 (48.3±0.4) | 93.7±0.1 (94.5±0.1) | 0.0±0.0 (16.1±2.4) | 36.0±0.8 |
| MaSS | 65.1±0.5 (83.7±0.2) | 49.8±0.2 (48.4±0.2) | 93.8±0.1 (94.6±0.0) | 0.0±0.0 (14.8±0.1) | 36.3±0.3 |
| PASS | 0.0±0.0 (57.0±0.0) | 0.0±0.0 (4.6±0.1) | 98.1±0.3 (96.8±0.2) | 97.6±0.3 (96.5±0.1) | **98.1±0.3** |

In addition to measuring the experiment results in NAG in Table 1 and Table 4, we also measure these experiment results in accuracy, as shown in Table 13 and Table 14 respectively.

To examine PASS's stochastic data substitution behavior closely, we visualize the results of data substitution using confusion matrices, as shown in Figure 4. We can observe that, for the useful attribute "activity", the diagonal values in the confusion matrix are significantly larger than the other values, which shows that most original samples are substituted by a sample with the same "activity" class. On the contrary, for private attributes "gender" and "ID", the rows and columns of the confusion matrix are highly independent, showing that original samples tend to be substituted by samples with highly random "gender" and "ID" classes. These results reveal the underlying logic of PASS's effectiveness.

We conduct another experiment to show that PASS can safely tolerate the Probing Attack even when the attacker has a larger dataset than the protector. In this experiment, we use the same setting as Table 14, except that the protector only has access to 50% of the training dataset to train PASS (and baselines), while the attacker has access to the entire training dataset to perform the Probing Attack. The results, as shown in Table 15, show that PASS achieves an mNAG that is still higher than all the baselines and is only 2.4% lower than when the protector has the entire training dataset.

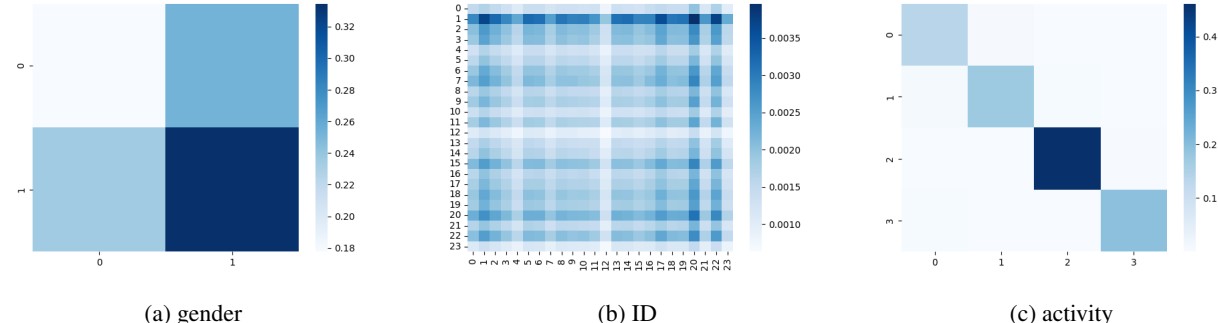

| (a) gender | (b) ID | (c) activity |

*Figure 4.* The confusion matrices of the stochastic data substitution results of PASS. The confusion matrices are calculated for the experiment shown in Table 4, where we suppress "gender", "id" as private attributes and preserve "activity" as a useful attribute. We report the confusion matrices for each attribute on the test set of Motion Sense. The value in the $i$-th row and $j$-th column of the confusion matrix represents the fraction of the original samples with the $i$-th class substituted by the substitute samples with the $j$-th class.

*Table 15.* Comparison of the NAG and accuracy between PASS and baselines on Motion Sense. We suppress "gender" and "ID" as private attributes, while preserving "activity" as useful attributes. NAG-unfinetuned means that this NAG is calculated with a classifier that is only pre-trained on original data but not finetuned on substituted data. In this experiment, the protector only has access to 50% of the training set, while the attacker has access to the entire training set.

| Method | NAG (Accuracy) (%) | | | mNAG (%) (↑) |
|---|---|---|---|---|
| | gender (↓) | ID (↓) | activity (↑) | |
| No suppr. | 100.0 (98.0) | 100.0 (91.7) | 100.0 (97.8) | 0.0 |
| Guessing | 0.0 (57.0) | 0.0 (5.3) | 0.0 (46.6) | 0.0 |
| ADV | 69.5±0.6 (85.5±0.2) | 47.3±0.3 (46.1±0.3) | 90.2±2.2 (92.7±1.1) | 31.8±1.9 |
| GAP | 68.0±0.7 (84.9±0.3) | 53.8±0.1 (51.7±0.1) | 87.3±0.2 (91.2±0.1) | 26.4±0.7 |
| MSDA | 64.3±1.6 (83.3±0.7) | 44.9±0.9 (44.1±0.8) | 88.5±0.4 (91.9±0.2) | 33.9±0.8 |
| BDQ | 67.2±2.2 (84.5±0.9) | 47.0±2.6 (45.9±2.2) | 90.9±0.5 (93.1±0.3) | 33.8±0.7 |
| PPDAR | 67.8±0.4 (84.8±0.2) | 51.4±1.2 (49.7±1.0) | 91.4±0.3 (93.3±0.2) | 31.8±0.4 |
| MaSS | 68.0±0.1 (84.9±0.1) | 53.1±0.1 (51.2±0.1) | 91.6±0.4 (93.5±0.2) | 31.0±0.4 |
| PASS | 0.0±0.0 (57.0±0.0) | 0.0±0.0 (4.7±0.1) | 95.7±0.2 (95.6±0.1) | **95.7±0.2** |

## F.3. Additional Results on CelebA

*Table 16.* Comparison of the NAG and accuracy between PASS and baselines on CelebA. We suppress "Male" as a private attribute, while preserving "Smiling" and "Young" as useful attributes, and we take "Attractive," "Mouth_Slightly_Open," and "High_Cheekbones" as hidden useful attributes to evaluate general feature preservation.

| Method | NAG (Accuracy) (%) | | | | | | mNAG (%) (↑) |
|---|---|---|---|---|---|---|---|
| | Male (↓) | Smiling (↑) | Young (↑) | Attractive (↑) | Mouth_Slightly_Open (↑) | High_Cheekbones (↑) | |
| No suppr. | 100.0 (98.9) | 100.0 (92.9) | 100.0 (87.9) | 100.0 (82.0) | 100.0 (94.0) | 100.0 (87.6) | 0.0 |
| Guessing | 0.0 (59.4) | 0.0 (50.8) | 0.0 (75.2) | 0.0 (50.8) | 0.0 (51.1) | 0.0 (53.4) | 0.0 |
| ADV | 99.9±0.1 (98.8±0.0) | 98.8±0.1 (92.5±0.0) | 97.0±0.9 (87.6±0.1) | 94.6±0.4 (80.3±0.1) | 99.1±0.1 (93.7±0.1) | 97.0±0.5 (86.6±0.2) | -2.6±0.2 |
| GAP | 83.0±1.1 (92.2±0.4) | 75.9±1.3 (82.8±0.5) | 45.4±3.0 (81.0±0.4) | 77.6±1.1 (75.0±0.4) | 61.1±2.1 (77.3±0.9) | 75.6±0.7 (79.3±0.2) | -15.9±2.3 |
| MSDA | 91.6±0.7 (95.5±0.3) | 99.8±0.2 (92.8±0.1) | 92.4±2.4 (87.0±0.3) | 89.9±1.0 (78.8±0.3) | 91.8±0.8 (90.5±0.4) | 95.7±1.1 (86.1±0.4) | 2.3±0.8 |
| BDQ | 99.7±0.1 (98.7±0.0) | 98.8±0.2 (92.4±0.1) | 96.3±0.8 (87.5±0.1) | 94.1±0.6 (80.2±0.2) | 98.9±0.4 (93.6±0.1) | 97.0±0.3 (86.6±0.1) | -2.7±0.2 |
| PPDAR | 99.7±0.1 (98.7±0.1) | 98.9±0.3 (92.5±0.1) | 97.2±1.2 (87.6±0.1) | 94.4±0.6 (80.2±0.2) | 99.0±0.1 (93.6±0.0) | 97.0±0.4 (86.6±0.1) | -2.4±0.3 |
| MaSS | 96.9±0.1 (97.7±0.0) | 97.2±0.2 (91.8±0.1) | 86.2±1.4 (86.2±0.2) | 90.6±0.3 (79.0±0.1) | 97.6±0.2 (93.0±0.1) | 94.6±0.4 (85.8±0.1) | -3.7±0.4 |
| PASS | 4.9±0.5 (61.3±0.2) | 98.3±0.1 (92.2±0.0) | 78.6±0.8 (85.2±0.1) | 58.1±2.8 (68.9±0.9) | 67.0±0.8 (79.9±0.3) | 86.7±0.3 (83.1±0.1) | **72.9±0.2** |

*Table 17.* Comparison of the NAG and accuracy between PASS and DP-based baselines on CelebA. We suppress "Male" as a private attribute, while preserving "Smiling" and "Young" as useful attributes, and we take "Attractive," "Mouth_Slightly_Open," and "High_Cheekbones" as hidden useful attributes to evaluate general feature preservation.

| Method | NAG (Accuracy) (%) | | | | | | mNAG (%) (↑) |
|---|---|---|---|---|---|---|---|
| | Male (↓) | Smiling (↑) | Young (↑) | Attractive (↑) | Mouth_Slightly_Open (↑) | High_Cheekbones (↑) | |
| No suppr. | 100.0 (98.9) | 100.0 (92.9) | 100.0 (87.9) | 100.0 (82.0) | 100.0 (94.0) | 100.0 (87.6) | 0.0 |
| Guessing | 0.0 (59.4) | 0.0 (50.8) | 0.0 (75.2) | 0.0 (50.8) | 0.0 (51.1) | 0.0 (53.4) | 0.0 |
| Snow | 97.8±0.1 (98.0±0.1) | 93.7±0.4 (91.3±0.1) | 91.9±0.5 (84.8±0.2) | 95.7±0.1 (91.1±0.0) | 80.4±1.2 (85.5±0.2) | 84.7±0.3 (77.2±0.1) | -8.5±0.3 |
| DPPix | 94.5±0.2 (96.7±0.1) | 81.7±0.2 (86.2±0.1) | 86.8±0.7 (83.1±0.2) | 91.4±0.2 (89.3±0.1) | 63.6±0.5 (83.3±0.1) | 78.1±0.5 (75.2±0.1) | -14.2±0.1 |
| Laplace Mechanism | 91.0±0.2 (95.3±0.1) | 79.3±0.1 (85.2±0.1) | 87.0±0.2 (83.2±0.1) | 89.8±0.5 (88.6±0.2) | 60.8±1.6 (83.0±0.2) | 81.2±0.4 (76.1±0.1) | -11.4±0.5 |
| DP-Image | 79.6±0.1 (90.8±0.1) | 68.5±0.2 (80.5±0.1) | 79.8±0.1 (80.7±0.1) | 79.8±0.2 (84.4±0.1) | 55.0±1.0 (82.2±0.1) | 78.7±0.6 (75.3±0.2) | -7.2±0.2 |
| PASS | 4.9±0.5 (61.3±0.2) | 98.3±0.1 (92.2±0.0) | 78.6±0.8 (85.2±0.1) | 58.1±2.8 (68.9±0.9) | 67.0±0.8 (79.9±0.3) | 86.7±0.3 (83.1±0.1) | **72.9±0.2** |

*Table 18.* Comparison of the NAG and accuracy between PASS and SPAct on CelebA. We suppress "Male" as a private attribute, while preserving "Smiling" and "Young" as useful attributes.

| Method | NAG (Accuracy) (%) | | | mNAG (%) (↑) |
|---|---|---|---|---|
| | Male (↓) | Smiling (↑) | Young (↑) | |
| No suppr. | 100.0 (98.9) | 100.0 (92.9) | 100.0 (87.9) | 0.0 |
| Guessing | 0.0 (59.4) | 0.0 (50.8) | 0.0 (75.2) | 0.0 |
| SPAct | 81.3±4.2 (91.5±1.7) | 94.0±1.9 (90.4±0.8) | 59.4±3.3 (82.8±0.4) | -4.6±2.7 |
| PASS | 4.9±0.5 (61.3±0.2) | 98.3±0.1 (92.2±0.0) | 78.6±0.8 (85.2±0.1) | 83.6±0.7 |

In addition to measuring the experiment results in NAG in Table 5, we also measure experiment results in accuracy, as shown in Table 16.

We further compared PASS with 4 additional DP-based baselines: Laplace Mechanism (Additive Noise) (Dwork et al., 2006), DPPix (Fan, 2018), Snow (John et al., 2020), and DP-Image (Xue et al., 2021). As shown in Table 17, these DP-based methods exhibit limited performance in the Private Attribute Protection task, because they focus on preventing the inference of membership from obfuscated samples, which is not fully aligned with our objective of preventing inference of specific private attributes from obfuscated samples while preserving utility.

Similar to the AudiMNIST dataset, we also conduct an experiment to show that PASS is robust to various combinations of private attributes and useful attributes. As shown in Table 19, PASS consistently achieves high mNAG for all combinations, even when the chosen private attributes and useful attributes are highly correlated (e.g., when we suppress "Smiling" but preserve "Mouth_Slightly_Open").

*Table 19.* Comparison of the NAG and accuracy of PASS on CelebA for different configurations. In each configuration, the attributes annotated with (S) are suppressed as private attributes, and the attributes annotated with (U) are preserved as useful attributes.

| Method | (suppressed (S) or preserved (U)) NAG (Accuracy) (%) | | | | | | mNAG (%) (↑) |
|---|---|---|---|---|---|---|---|
| | Male | Smiling | Young | Attractive | Mouth_Slightly_Open | High_Cheekbones | |
| No suppr. | (U) 100.0 (98.9) | (U) 100.0 (92.9) | (U) 100.0 (87.9) | (U) 100.0 (82.0) | (U) 100.0 (94.0) | (U) 100.0 (87.6) | 0.0 |
| Guessing | (S) 0.0 (59.4) | (S) 0.0 (50.8) | (S) 0.0 (75.2) | (S) 0.0 (50.8) | (S) 0.0 (51.1) | (S) 0.0 (53.4) | 0.0 |
| PASS | (S) 3.6±0.5 (60.8±0.2) | (U) 96.7±0.2 (91.5±0.1) | (U) 63.5±1.9 (83.3±0.2) | (U) 73.1±0.6 (73.6±0.2) | (U) 97.9±0.3 (93.1±0.1) | (U) 89.7±0.3 (84.1±0.1) | 80.6±0.5 |
| | (S) 9.8±0.8 (63.2±0.3) | (S) 45.6±0.7 (70.0±0.3) | (U) 81.6±1.7 (85.6±0.2) | (U) 80.7±0.9 (76.0±0.3) | (U) 80.4±1.0 (85.7±0.4) | (U) 53.2±1.3 (71.6±0.4) | 46.3±0.4 |
| | (S) 13.9±0.3 (64.9±0.1) | (S) 61.8±0.7 (76.9±0.3) | (S) 0.0±0.0 (75.2±0.0) | (U) 73.9±0.3 (73.9±0.1) | (U) 94.0±0.2 (91.5±0.1) | (U) 74.5±0.7 (78.9±0.2) | 55.6±0.1 |
| | (S) 5.6±0.5 (61.6±0.2) | (S) 72.1±0.9 (81.2±0.4) | (S) 0.0±0.0 (75.2±0.0) | (S) 14.4±0.3 (55.3±0.1) | (U) 97.1±0.2 (92.8±0.1) | (U) 86.6±0.4 (83.0±0.1) | 68.8±0.0 |
| | (S) 6.8±0.4 (62.1±0.1) | (S) 77.2±0.9 (83.3±0.4) | (S) 0.0±0.0 (75.2±0.0) | (S) 16.8±0.3 (56.0±0.1) | (S) 45.6±0.7 (70.7±0.3) | (U) 93.7±0.8 (85.4±0.3) | 64.4±0.4 |

Apart from these experiments, we also compare PASS with SPAct(Dave et al., 2022) on CelebA. However, since SPAct proposes to suppress general features while PASS aims at preserving general features, their performances on general features are not comparable. Therefore, we only compare their performance on private attributes and useful attributes. In a task suppressing the "Male" as private attribute while preserving "Smiling", "Young" as useful attributes, we can observe from the results shown in Table 18, that PASS achieves significantly higher mNAG than SPAct, substantiating PASS's effectiveness.

# G. Discussion on Scalability

While our proposed method PASS demonstrates strong performance in private attribute protection, it shares a common limitation with existing state-of-the-art methods: the need for retraining when the set of private attributes changes. However, compared to other baseline approaches, PASS offers a notable advantage in training efficiency, as its loss function is computed through embedding extraction followed by a cosine similarity operation (please see Section 4.3 for details).

It is also worth noting that although certain DP-based methods do not require retraining, their focus on broader membership protection inherently limits their effectiveness for the specific task of private attribute protection, as demonstrated in Table 17 above.

