# OpenReview forum: "PASS: Private Attributes Protection with Stochastic Data Substitution"
_ICML.cc/2025/Conference — ICML 2025 spotlightposter_

### Official Review · Reviewer_aH9x · 2025-03-05

**Overall Recommendation:** 4

**Summary:**

This paper introduces PASS (Private Attributes protection with Stochastic data Substitution), a novel approach to protect private attributes in user data while preserving utility. Unlike existing adversarial training-based methods, PASS employs stochastic data substitution where each original sample is replaced with a sample from a substitute dataset according to learned probabilities. The method is derived from an information-theoretic framework and demonstrates superior robustness against probing attacks across multiple data modalities (images, sensor signals, speech).

**Claims And Evidence:**

Yes

**Essential References Not Discussed:**

No

**Experimental Designs Or Analyses:**

The experimental design is sound, comparing PASS against six state-of-the-art methods across three datasets. The ablation studies with varying hyperparameters demonstrate robustness. The asymmetric scenario (Table 13) where attackers have more data than protectors is a particularly strong validation. I reviewed all experimental sections, including the appendices showing additional results.

**Methods And Evaluation Criteria:**

Yes

**Other Comments Or Suggestions:**

No

**Other Strengths And Weaknesses:**

Weaknesses:
1.The paper overlooks memory requirements for storing substitute datasets and embeddings, particularly problematic for high-dimensional data applications.
2.Critical visualizations comparing original and substituted samples are missing, which would better demonstrate the method's effectiveness.
3.The substitute dataset construction relies on random sampling without justification or exploration of more optimal selection strategies.

**Questions For Authors:**

No

**Relation To Broader Scientific Literature:**

PASS extends privacy protection literature by connecting local differential privacy with utility-preserving attribute protection. It builds upon information bottleneck/privacy funnel concepts (Makhdoumi et al., 2014; Alemi et al., 2016) but applies them in a novel way through data substitution rather than feature transformation. The work also relates to k-anonymity and l-diversity literature by ensuring multiple samples with different private attributes become indistinguishable.

**Theoretical Claims:**

I examined the proofs for Theorem 4.1 (Appendix D.2) connecting the approximate loss Ĺ to the true objective L, and Theorem 4.2 (Appendix A) analyzing entangled attributes. Both proofs are mathematically sound, correctly applying information theory principles to establish bounds and relationships between mutual information terms. No issues were identified in the derivations.

---

> ### Author Rebuttal · Authors · 2025-04-01
>
> **Q1**. The paper overlooks memory requirements for storing substitute datasets and embeddings, particularly problematic for high-dimensional data applications.
>
> **A1**. Thanks for your question! Importantly, the substitution dataset itself does not need to be loaded into memory during inference. Instead, we only need to load the corresponding embeddings into memory to calculate the substitution probability $P_\theta(X'|X)$. which greatly reduces memory usage.
>
> Furthermore, as demonstrated in the ablation study on substitution dataset size (Appendix F.1, Table 10), PASS maintains consistently strong performance across a wide range of substitution dataset sizes. This flexibility allows users to adapt the size of the substitution dataset based on their available storage and memory. For instance, if a user has limited secondary storage to store the substitution dataset, or limited memory to load its embeddings, then they can opt to use a smaller substitution dataset without substantial loss in performance.
>
> In addition, users can further control memory usage by adjusting the dimension of the embeddings. Users with limited memory can reduce the embedding dimension to fit their resource constraints.
>
> These characteristics allow PASS to be adapted to different hardware settings without compromising its core functionality.
>
>
> **Q2**. Critical visualizations comparing original and substituted samples are missing, which would better demonstrate the method's effectiveness.
>
> **A2**. Thanks for the insightful suggestion! We will visualize the original and the substituted samples in our revised paper to help readers understand PASS's behavior and showcase PASS's effectiveness.
>
>
> **Q3**. The substitute dataset construction relies on random sampling without justification or exploration of more optimal selection strategies.
>
> **A3**. Thanks for the thoughtful question! To better understand how the choice of substitution dataset affects PASS’s performance, we conducted a new ablation study on the AudioMNIST dataset. In this study, we constructed multiple substitution datasets with varying attribute distributions:
>
> 1. For the sensitive attribute "gender", we varied its distribution from "90\% Male / 10\% Female" to "10\% Male / 90\% Female".
>
> 2. For the useful attribute "digit", we varied its distribution from uniform distribution "10\% 0-9" to a highly skewed distribution "50\% 0 / 5.6\% 1-9".
>
> The results, shown in **Table A** below, demonstrate that PASS consistently maintains high performance across all these different substitution dataset configurations, even under highly imbalanced conditions. This ablation study provides empirical evidence that the specific selection of the substitution dataset has a limited impact on PASS's overall performance. Consequently, it may justify that a simple random sampling strategy is sufficient to construct an effective substitution dataset and achieve strong PASS performance.
>
> That said, we agree that there may still be room to further enhance PASS's effectiveness with more advanced substitution dataset selection strategies. We plan to explore this direction in our future work.
>
>
>
>
> **Table A**. Ablation study on the distribution of substitution dataset. The other settings are the same as Table 2. Results are averaged over 3 random seeds (STD is omitted to save space).
>
> |gender distribution|digit distribution|gender NAG(↓)|accent NAG(↑)|age NAG(↑)|ID NAG(↑)|digit NAG(↑)|mNAG(↑)|
> |-|-|-|-|-|-|-|-|
> |90% Male / 10% Female|10% 0-9|0.0|47.9|28.4|51.2|96.9|56.1|
> |80% Male / 20% Female|10% 0-9|0.0|46.4|27.6|49.7|96.5|55.0|
> |50% Male / 50% Female|10% 0-9|0.1|47.0|28.1|50.5|96.5|55.4|
> |20% Male / 80% Female|10% 0-9|0.0|47.4|27.8|50.8|96.7|55.6|
> |10% Male / 90% Female|10% 0-9|0.0|47.8|27.4|51.2|96.3|55.7|
> |80% Male / 20% Female|30% 0 / 7.8% 1-9|0.0|46.7|27.6|50.0|96.0|55.1|
> |80% Male / 20% Female|50% 0 / 5.6% 1-9|0.0|44.3|25.3|48.1|93.9|52.9|

---

### Official Review · Reviewer_Acbi · 2025-03-13

**Overall Recommendation:** 4

**Summary:**

PASS (Private Attributes Protection with Stochastic Data Substitution) introduces a novel method to protect private attributes in machine learning datasets by replacing original data samples with others from a substitution dataset using a stochastic algorithm trained with an information-theoretic loss. Unlike existing adversarial training-based methods, which are vulnerable to probing attacks, PASS provides a stronger defense by offering a theoretical foundation for balancing privacy and utility. Empirical results on datasets like CelebA, Motion Sense, and AudioMNIST show that PASS effectively protects private attributes while preserving data utility.

**Claims And Evidence:**

The claims made in this submission are mostly supported by empirical and theoretical evidence. However, there are still one concerning point: while PASS is compared to k-l-t privacy and LDP, it lacks direct benchmarks against modern differential privacy mechanisms such as DP-SGD. This makes the privacy guarantees of PASS less quantifiable and more difficult to evaluate in comparison to formal differential privacy methods.

The submission presents ablation studies (Tables 9, 10) demonstrating robustness to $\mu$ and $|\mathcal{D}_{\text{substitute}}|$. However, it does not provide clear guidelines for selecting $\lambda$ and $\mu$ in practice, which could make it difficult for users to balance privacy-utility trade-offs without relying on trial-and-error tuning. Additionally, w

**Essential References Not Discussed:**

As far as I know, no related works that are essential to understanding the (context for) key contributions of the paper, but are not currently cited/discussed in the paper.

**Experimental Designs Or Analyses:**

1. In the Motion Sense experiments, the authors evaluate the performance of an "unfinetuned" classifier (a classifier only pre-trained on the original data without fine-tuning on the substituted data). While this provides some insight into the transferability of features, it's not clear why this metric is only used for the Motion Sense dataset and not the others.

2. The paper states that hyperparameters are set to balance privacy protection, utility preservation, and general feature preservation. However, the specific process of hyperparameter tuning is not described in detail. This raises the possibility that the performance of PASS could be sensitive to the choice of hyperparameters.

**Methods And Evaluation Criteria:**

The paper primarily uses Normalized Accuracy Gain (NAG) and mean NAG (mNAG) as evaluation metrics. While NAG appropriately accounts for class imbalance, it depends on classifier accuracy, which could conflate the quality of obfuscation with classifier robustness. Incorporating additional metrics such as precision, recall, and F1-score could provide a more comprehensive and balanced evaluation of the method's performance.

**Other Comments Or Suggestions:**

1. It may be helpful to apply the "unfinetuned" classifier metric to other datasets for consistency and broader insights into transferability.
2. Providing more details on the hyperparameter tuning process would help clarify how privacy, utility, and feature preservation are balanced.
3. Adding direct benchmarks against modern DP methods (e.g., DP-SGD) could make the privacy guarantees more comparable and easier to quantify.
4. Discussing potential limitations of the stochastic data substitution approach, particularly in high-dimensional or highly imbalanced data, would strengthen the analysis.

**Other Strengths And Weaknesses:**

Strengths

1. PASS introduces a creative approach to private attribute protection by using stochastic data substitution instead of adversarial training. This represents an original combination of ideas from randomized response techniques and information theory.

2. The method is rigorously grounded in information theory, with clear derivations and proofs connecting the practical loss function to the theoretical objectives. This provides a solid mathematical basis for the approach.

3. PASS demonstrates effectiveness across multiple data modalities (audio, sensor data, images), suggesting it can generalize well to different types of private attribute protection tasks.

4. By addressing the vulnerability of existing methods to probing attacks, PASS tackles a real-world concern in deploying privacy-preserving machine learning systems.

Weaknesses

1. While PASS is compared to several baselines, the evaluation lacks comparison to differential privacy methods, which are considered the gold standard for privacy guarantees.

2. Although PASS shows empirical privacy protection, it does not provide formal privacy guarantees comparable to differential privacy's $\epsilon$-$\delta$ bounds. This makes it challenging to precisely quantify the level of privacy achieved.

3. While some ablation studies are provided, there is limited guidance on selecting optimal hyperparameters (λ, μ) for balancing privacy and utility in practice.

**Questions For Authors:**

1. Could you clarify why the "unfinetuned" classifier metric was only applied to the Motion Sense dataset and not to other datasets? Applying it more broadly could provide a more comprehensive view of feature transferability.

2. Could you provide more details on how the hyperparameters (e.g., $\lambda$ and $\mu$) were tuned? Understanding this process would clarify how privacy, utility, and feature preservation are balanced and help assess the generalizability of the approach.

3. Do you plan to include direct benchmarks against modern DP methods such as DP-SGD? This would make it easier to compare PASS’s privacy guarantees with those of established differential privacy techniques.

**Relation To Broader Scientific Literature:**

The paper introduces a novel stochastic data substitution approach that avoids adversarial training altogether. This method relates to broader research on randomized response techniques in privacy-preserving data analysis. However, PASS extends these concepts to high-dimensional data spaces and incorporates utility-preserving requirements, distinguishing it from traditional randomized response methods.

**Theoretical Claims:**

I have verified the theoretical claims and proofs presented in the paper.

---

> ### Author Rebuttal · Authors · 2025-04-01
>
> **Q1**: "...apply the "unfinetuned" classifier metric to other datasets..."
>
> **A1**: Thanks for the suggestion! We applied the "NAG-unfinetuned" metric to the useful attributes in the AudioMNIST and CelebA datasets for consistency. The results are shown in **Tables A** and **B**, respectively. For unfinetuned classifiers, PASS still offers the best balance between privacy protection and utility preservation. This is due to our loss function, which encourages replacing each sample with others sharing the same useful attributes. That said, we acknowledge that fine-tuning downstream classifiers can further improve performance, and we plan to explicitly explore adapting PASS to better suit unfinetuned classifiers in future work.
>
>
> **Table A**. Comparison with baselines, using "NAG-unfinetuned" metric on useful attributes. The settings are the same as Table 2. NAG on the private attribute "gender" is also presented for reference. Results are averaged over 3 random seeds (STD is omitted to save space).
>
> |Method|gender NAG(↓)|digit NAG-unfinetuned(↑)|
> |-|-|-|
> |ADV|71.4|99.7|
> |GAP|13.3|0.3|
> |MSDA|78.4|0.0|
> |BDQ|69.0|5.9|
> |PPDAR|81.7|66.2|
> |MASS|88.9|31.4|
> |PASS|0.0|77.0|
>
>
> **Table B**. Comparison with baselines, using "NAG-unfinetuned" metric on useful attributes. The settings are the same as Table 5. NAG on the private attribute "Male" is also presented for reference. Results are averaged over 3 random seeds (STD is omitted to save space).
>
> |Method|Male NAG(↓)|Smiling NAG-unfinetuned(↑)|Young NAG-unfinetuned(↑)|
> |-|-|-|-|
> |ADV|99.9|99.9|98.3|
> |GAP|83.0|0.0|0.0|
> |MSDA|91.6|0.0|0.0|
> |BDQ|99.7|98.5|90.2|
> |PPDAR|99.7|98.1|92.7|
> |MASS|96.9|0.0|0.0|
> |PASS|4.9|85.2|49.1|
>
>
> **Q2**. "...guidance on selecting optimal hyperparameters ($\lambda$, $\mu$)..."
>
> **A2**. To guide the selection of $\lambda$ and $\mu$, we conducted the following ablation studies. For $\lambda$, the results in **Table C** below show that PASS consistently achieves near-0 NAG on private attribute and stable overall performance across a wide range of values. Thus, we recommend the users to simply fix $\lambda=N/M$ without extensive tuning, where $N$ and $M$ are numbers of useful and private attributes, respectively. This choice is used throughout this paper. For $\mu$, results in Appendix F.1 Table 9 show that increasing $\mu$ enhances general feature preservation but may slightly reduce useful attribute preservation. Importantly, privacy protection remains strong across all $\mu$. Therefore, we suggest that $\mu$ can be flexibly adjusted based on the relative importance of general features in users' specific tasks, without compromising privacy protection.
>
>
> **Table C**. Ablation study on $\lambda$. The other settings are the same as Table 2. Results are averaged over 3 random seeds (STD is omitted to save space).
>
> |$\lambda$|gender NAG(↓)|accent NAG(↑)|age NAG(↑)|ID NAG(↑)|digit NAG(↑)|mNAG(↑)|
> |-|-|-|-|-|-|-|
> |0.1N/M|0.0|45.0|25.2|48.5|88.5|51.8|
> |0.2N/M|0.0|45.3|25.7|49.3|91.5|53.0|
> |0.5N/M|0.0|47.3|28.4|50.8|95.0|55.4|
> |1N/M|0.0|46.4|27.6|49.7|96.5|55.0|
> |2N/M|0.0|47.8|28.5|51.3|97.8|56.4|
> |5N/M|0.1|46.2|27.0|49.9|99.0|55.4|
>
> **Q3**. "...direct benchmarks against modern DP methods (e.g., DP-SGD)..."
>
> **A3**. We compared PASS with four additional DP-based baselines: Laplace Mechanism (Additive Noise) [1], DPPix [2], Snow [3], and DP-Image [4]. Due to space limitations, the results are presented in **Table A** in our response to **Reviewer RCzr** above. These methods show limited effectiveness on the Private Attribute Protection task because they are designed to prevent inference of **membership** from obfuscated samples, which is not fully aligned with our objective of preventing inference of **specific private attributes** from obfuscated samples while preserving the utility.
>
> In contrast, DP-SGD [5] aims to protect against membership inference **from a model’s trained parameters**, which is orthogonal to our goal—PASS focuses on preventing private attributes inference **from obfuscated samples**. Therefore, DP-SGD is not directly comparable to PASS, but could potentially be combined with PASS to provide more comprehensive privacy protection.
>
>
> References [1-4] are shown in our response **A3** to **Reviewer RCzr** due to space limitations.
>
> [5]: Abadi, Martin, et al. "Deep learning with differential privacy." Proceedings of the 2016 ACM SIGSAC conference on computer and communications security. 2016.
>
>
> **Q4**. "Discussing potential limitations... particularly in high-dimensional or highly imbalanced data..."
>
> **A4**. Thanks for the suggestion! While we have demonstrated PASS's effectiveness on moderately high-dimensional data (e.g., 3×160×160 images in CelebA) and imbalanced datasets (e.g., 80\% Male / 20\% Female for "gender" in AudioMNIST), we acknowledge that handling even higher-dimensional or more imbalanced data may demand PASS to learn more representative features per sample, posing intrivial challenges to PASS's robustness and scalability.

---

> > ### Comment · Reviewer_Acbi · 2025-04-06
> >
> > Thank you for the detailed rebuttal. The authors have addressed the key concerns well. Overall, the rebuttal significantly improves the submission. I’m upgrading my score.

---

> > > ### Author Response · Authors · 2025-04-06
> > >
> > > Thanks for your comments. We genuinely appreciate your careful review and are grateful that our responses helped clarify the key concerns!

---

### Official Review · Reviewer_f4yf · 2025-03-14

**Overall Recommendation:** 3

**Summary:**

This paper addresses the challenge of protecting private attributes in machine learning (ML) services while preserving the utility of the data for downstream tasks. Existing methods primarily rely on adversarial training to remove private attributes, but the authors identify a fundamental vulnerability in these approaches, both theoretically and empirically.

To mitigate this issue, the paper introduces PASS, a novel stochastic substitution mechanism that replaces original samples with alternative ones based on a probabilistic framework. The approach is guided by a newly designed loss function, rigorously derived from an information-theoretic objective. Extensive experiments across multiple modalities—facial images, human activity sensor data, and voice recordings—demonstrate PASS's effectiveness and generalizability.

The proposed method offers a fresh perspective on privacy-preserving ML, particularly by moving away from adversarial training. The work is well-motivated, and the empirical results strengthen its claims.

**Claims And Evidence:**

Yes, the claims made in the submission are supported by clear and convincing evidence.

**Essential References Not Discussed:**

Not sure.

**Experimental Designs Or Analyses:**

Yes, I have checked all the experiments and it looks sound to me.

**Methods And Evaluation Criteria:**

Yes.

**Other Comments Or Suggestions:**

Typo: Line 32: image/(vedio) detection.

**Other Strengths And Weaknesses:**

Strengths:
1. The paper is very well-written and easy to follow. Even for a non-expert, the key concepts are clearly conveyed.
Weaknesses:
1. The experiments consider only a single private attribute per dataset, despite multiple useful attributes being present. Evaluating the method with multiple private attributes would strengthen the analysis.

2. The NAG metric lacks a clear explanation. More details are needed on its computation, particularly how Acc_guessing is derived.

**Questions For Authors:**

What happens when we use other attributes like age is used as private?

**Relation To Broader Scientific Literature:**

This work significantly advances privacy-preserving ML by identifying fundamental weaknesses in adversarial training approaches and proposing PASS, a theoretically grounded and empirically validated alternative. By leveraging probabilistic sample substitution rather than adversarial representation learning, the method contributes to both the privacy literature and broader discussions on fairness and information-theoretic ML.

**Theoretical Claims:**

I have checked proof of theorem 4.2 and section D in the appendix. It seems correct.

---

> ### Author Rebuttal · Authors · 2025-04-01
>
> **Q1**: "The experiments consider only a single private attribute per dataset, despite multiple useful attributes being present. Evaluating the method with multiple private attributes would strengthen the analysis."
>
> **A1**: Thanks for your question. This paper included experiments with multiple private attributes to demonstrate PASS’s scalability.
> - In Table 3, we experimented with up to 4 private attributes: "gender", "accent", "age", and "ID", on the AudioMNIST dataset.
> - In Tables 4 and 13, we experimented with 2 private attributes: "gender" and "ID", on the MotionSense dataset.
> - In Table 16, we experimented with up to 5 private attributes: "Male", "Smiling", "Young", "Attractive", and "Mouth\_Slightly\_Open", on the CelebA dataset.
>
> Across all settings, PASS consistently delivered a strong performance, highlighting its ability to scale to varying numbers and combinations of private attributes. Please see Sections 5.2, 5.3, Appendices F.2 and F.3 for details.
>
>
> **Q2**: "The NAG metric lacks a clear explanation. More details are needed on its computation, particularly how Acc\_guessing is derived."
>
> **A2**: Thanks for pointing this out. We will include a more detailed explanation of NAG below as an extension to Section 5.1. NAG, proposed by Chen et al. [1], is a metric for evaluating Private Attribute Protection methods.  For private attribute $S_i$, the NAG  is defined as
> $$
>     NAG(S_i) = \max\left(0,\frac{Acc(S_i) - Acc_\text{guessing}(S_i)}{Acc_\text{no-suppr.}(S_i) - Acc_\text{guessing}(S_i)}\right),
> $$
> where
> - $Acc(S_i)$ is the accuracy of a classifier trained to classify $S_i$ from the substituted sample $X'$.
> - $Acc_\text{guessing}(S_i)$ is the accuracy of a majority classifier (a classifier that always predicts the most frequent class regardless of the input, i.e., a poor classifier that can only "guess"), which in most cases represents the lower bound performance of $Acc(S_i)$.
> - $Acc_\text{no-suppr.}(S_i)$ is the classification accuracy of a classifier trained to classify $S_i$ from the original data $X$, which in most cases represents the upper bound performance of $Acc(S_i)$.
>
> By normalizing $Acc(S_i)$ between the lower bound ($Acc_\text{guessing}(S_i)$) and the upper bound ($Acc_\text{no-suppr.}(S_i)$), we ensure that NAG is on a consistent scale across all attributes, regardless of whether an attribute is balanced or imbalanced, or whether it is easy or difficult to predict. As a result, NAG provides a fair and reliable measure of how well each private attribute is suppressed or preserved. This makes it a suitable and consistent metric for comparing the effectiveness of different Private Attribute Protection methods. We will include this detailed description of NAG in our revised paper.
>
> [1]: Chen, Yizhuo, et al. "MaSS: multi-attribute selective suppression for utility-preserving data transformation from an information-theoretic perspective." Proceedings of the 41st International Conference on Machine Learning. 2024.
>
> **Q3**: "What happens when we use other attributes like age is used as private?"
>
> **A3**: Thanks for your question. This paper included experiments using "age" as a private attribute on the AudioMNIST dataset, as shown in Section 5.2, Table 3. Similarly, we experimented with "Young" as a private attribute on the CelebA dataset, detailed in Appendix F.2, Table 16. In both cases, PASS demonstrated a robust ability to suppress the private attribute "age" or "Young", while effectively preserving other useful attributes.
>
> Throughout the paper, we have evaluated PASS on a wide range of private attributes, including "gender", "accent", "age", "ID", "Male", "Smiling", "Young", "Attractive", and "Mouth\_Slightly\_Open". Across all these scenarios, PASS consistently achieved strong performance, reinforcing its broad applicability and versatility in handling diverse Private Attribute Protection tasks.

---

### Official Review · Reviewer_RCzr · 2025-03-18

**Overall Recommendation:** 3

**Summary:**

This paper proposes a feature substitution method based on an information-theoretic objective to preserve privacy for certain data attributes. The method does not depend on any specific adversarial strategy, making it more robustness than existing adversarial-based approaches.

#update after rebuttal:  the rebuttal has addressed most of the my concerns, except that I still think an end-to-end algorithm is much needed to make the approach easy to understand, and the limitations on scalability should be discussed in the final version. Overall I raised my rating and I hope the final version would incorporate these suggestions.

**Claims And Evidence:**

The claims are supported by both theoretical analysis and experimental evidence.

**Essential References Not Discussed:**

No.

**Experimental Designs Or Analyses:**

The experiments include several datasets and baselines. However，the baseline methods are all adversarial-based. The paper should also compare with other general defenses, such as DP,  adding noise, sparsification.

It is also suggested that the experiments include comprehensive evaluations on key impacting factors. For example, the distribution and quantity of a substitution dataset seems to be a key factor for the method to work well. However, it is not fully discussed how the composition of dataset affect the performance.

Another issue is that the method may be difficult to scale on features. It seems that adding additional features will involve retrain the entire model, which may be a challenge compared to DP.   The computation cost regarding training should also be presented.

**Methods And Evaluation Criteria:**

Key technical details are missing so it is hard to assess the soundness of the approach. Specifically, it is unclear from the paper how is this PASS method trained end-to-end. A formal algorithm depicting the steps is strongly suggested. Additional, there are missing details regarding:
1. Embedding function g(x'). The paper states that it is a "learnable" embedding, but it seems that no trainable parameters are assigned. It is unclear how this embedding function is performed or chosen.
2. Sample replacement vs. NN training. It is not clear how is the sample replacement step integrated with the NN training. e.g. Are all the samples replaced in the batch training? how often are parameters updated for each time that one or some samples are replaced?

**Other Comments Or Suggestions:**

No.

**Other Strengths And Weaknesses:**

I am a bit concerned that the boarder applicability of the proposed PASS approach maybe limited by its scalability on features or availability of abundant substitution dataset.  I suggest the paper discusses these aspects in more details.

**Questions For Authors:**

See the above comments.

**Relation To Broader Scientific Literature:**

This paper builds upon the existing literature on privacy protection of sensitive attributes, offering new insights from an information-theoretic approach.

**Theoretical Claims:**

No I didn't check the correctness of the proof.  However, one issue is that it is unclear how this method is compared with DP, both theoretically and experimentally.  The theoretical connection with DP seems to show that these are equivalent approaches  under certain assumptions, but it does not demonstrate that the approach is advantageous compared to DP.  It is suggested that the paper includes more discussions on the comparison with DP.

---

> ### Author Rebuttal · Authors · 2025-04-01
>
> **Q1**: "Embedding function $g(x')$."
>
> **A1**: Thanks for pointing it out! Our embedding function $g(x')$ is implemented as an embedding layer with trainable parameters (e.g., like the embedding layer in language models). We will update it as $g_\psi(x')$ to indicate its associated parameters clearly.
>
> **Q2**: "Sample replacement vs. NN training."
>
> **A2**: We do not need to perform "sample replacement" during training. Because our novel loss function (Equation 3) can be calculated with solely the substitution probability $P_\theta(X'|X)$, represented as a matrix for each batch, reducing training computation costs.
>
> **Q3**: "The paper should also compare with other general defenses, such as DP..."
>
> **A3**: Theoretically, as shown in Appendix B, PASS can be viewed as a Local DP method—specifically, a generalized form of Randomized Response with desirable DP properties.
>
> Experimentally, as suggested by the Reviewer, we further compared PASS with 4 additional DP-based baselines: Laplace Mechanism (Additive Noise) [1], DPPix [2], Snow [3], and DP-Image [4]. As shown in **Table A** below, these DP methods exhibit limited performance in the Private Attribute Protection task, because they focus on preventing the inference of **membership** from obfuscated samples, which is not fully aligned with our objective of preventing inference of **specific private attributes** while preserving utility.
>
> **Table A**. Comparison with DP-based baselines. The other settings are the same as Table 5. Results are averaged over 3 random seeds (STD is omitted to save space).
>
> |Method|Male NAG(↓)|Smiling NAG(↑)|Young NAG(↑)|Attractive NAG(↑)|Mouth\_Slightly\_Open NAG(↑)|High\_Cheekbones NAG(↑)|mNAG(↑)|
> |-|-|-|-|-|-|-|-|
> |SNOW|97.8|93.7|91.9|95.7|80.4|84.7|-8.5|
> |DPPix|94.5|81.7|86.8|91.4|63.6|78.1|-14.2|
> |Laplace Mechanism|91.0|79.3|87.0|89.8|60.8|81.2|-11.4|
> |DP-Image|79.6|68.5|79.8|79.8|55.0|78.7|-7.2|
> |PASS|4.9|98.3|78.6|58.1|67.0|86.7|**72.9**|
>
> [1]: Dwork et al. "Calibrating noise to sensitivity in private data analysis." Theory of Cryptography: Third Theory of Cryptography Conference, TCC. 2006.
>
> [2]: Fan, Liyue. "Image pixelization with differential privacy." IFIP Annual Conference on Data and Applications Security and Privacy. 2018.
>
> [3]: John et al. "Let it snow: Adding pixel noise to protect the user’s identity." ACM Symposium on Eye Tracking Research and Applications. 2020.
>
> [4]: Xue et al. "Dp-image: Differential privacy for image data in feature space." arXiv preprint. 2021.
>
>
> **Q4**: "...include comprehensive evaluations on key impacting factors. For example, the distribution and quantity of a substitution dataset..."
>
> **A4**: Thanks for the suggestion! Below, we demonstrate PASS's strong stability across varying quantities and distributions of substitution datasets. First, an ablation study on substitution dataset **quantity** (Appendix F.1, Table 10) shows that PASS maintains high performance across a wide range of substitution dataset sizes (1024 to 24,000). Second, we conducted a new ablation study on the **distribution** of the substitution dataset on AudioMNIST, where we varied the distribution of:
> 1. the sensitive attribute "gender", from "90\% Male / 10\% Female" to "10\% Male / 90\% Female".
> 2. the useful attribute "digit" from "10\% 0-9" to "50\% 0 / 5.6\% 1-9".
>
> As shown in **Table B**, PASS achieved consistently high performance, even on highly skewed distributions.
>
> **Table B**. Ablation study on the distribution of substitution dataset. The other settings are the same as Table 2. Results are averaged over 3 random seeds (STD is omitted to save space).
>
> |gender distribution|digit distribution|gender NAG(↓)|accent NAG(↑)|age NAG(↑)|ID NAG(↑)|digit NAG(↑)|mNAG(↑)|
> |-|-|-|-|-|-|-|-|
> |90% Male / 10% Female|10% 0-9|0.0|47.9|28.4|51.2|96.9|56.1|
> |80% Male / 20% Female|10% 0-9|0.0|46.4|27.6|49.7|96.5|55.0|
> |50% Male / 50% Female|10% 0-9|0.1|47.0|28.1|50.5|96.5|55.4|
> |20% Male / 80% Female|10% 0-9|0.0|47.4|27.8|50.8|96.7|55.6|
> |10% Male / 90% Female|10% 0-9|0.0|47.8|27.4|51.2|96.3|55.7|
> |80% Male / 20% Female|30% 0 / 7.8% 1-9|0.0|46.7|27.6|50.0|96.0|55.1|
> |80% Male / 20% Female|50% 0 / 5.6% 1-9|0.0|44.3|25.3|48.1|93.9|52.9|
>
> **Q5**: "...the method may be difficult to scale on features..."
>
> **A5**: All baseline state-of-the-art Private Attribute Protection methods in this paper require retraining when private attributes change. In comparison, PASS has a lower training cost than them, as its loss can be computed without actual sample replacement (please see **A2** and the last paragraph of Section 4.1 for more details).
>
> Furthermore, PASS is designed with scalability in mind, as it supports any number of private and useful attributes without substantially increasing computational cost.
>
> In addition, although some DP-based methods do not require retraining, their focus on general **membership protection** unavoidably limits their effectiveness on the Private Attributes Protection task, as shown in **A3** above.

---

### Decision · Program_Chairs · 2025-05-01

**Decision:**

Accept (spotlight poster)

**Comment:**

The paper introduces PASS (Private Attributes protection with Stochastic data Substitution) for protecting private attributes in machine learning datasets while preserving utility for downstream tasks. The authors argue that existing adversarially training methods are fundamentally vulnerable to probing attacks. In contrast, PASS leverages a stochastic substitution mechanism guided by an information-theoretic objective. This mechanism probabilistically replaces original data samples with alternative ones drawn from a substitution dataset. Theoretical analysis and comprehensive experiments across diverse datasets (including images, sensor signals, and speech) are performed to demonstrate the effectiveness and generalizability of the proposed method.


Several reviewers appreciated the paper’s well-founded theoretical framework, particularly its derivation from information-theoretic principles. They also appreciated the avoidance of adversarial training approaches (unlike existing works). The reviewers also highlighted: the strength of the empirical evaluations, clarity of writing, and the inclusion of thorough appendices. However, multiple reviewers raised concerns about the lack of comparisons with differential privacy (DP) techniques, especially DP-SGD, which would have provided a stronger baseline for quantifying privacy guarantees. In response, the authors argue that DP-SGD purpose is different (and it aims to defend against MIAs). I personally agree with this point of the authors. Additionally, there were shared concerns about scalability and practicality. Specifically, the potential computational cost and memory demands.  While the authors responded that only embeddings need to be loaded in the memory, they still did not provide clear memory/computational complexity result in their response.